# Towards Practical Defect-Focused Automated Code Review

**Junyi Lu** [1 2 †]  **Lili Jiang** [3]  **Xiaojia Li** [3]  **Jianbing Fang** [4 †]  **Fengjun Zhang** [1]  **Li Yang** [1]  **Chun Zuo** [5]

## Abstract

The complexity of code reviews has driven efforts to automate review comments, but prior approaches oversimplify this task by treating it as snippet-level code-to-text generation and relying on text similarity metrics like BLEU for evaluation. These methods overlook repository context, real-world merge request evaluation, and defect detection, limiting their practicality. To address these issues, we explore the full automation pipeline within the online recommendation service of a company with nearly 400 million daily active users, analyzing industry-grade C++ codebases comprising hundreds of thousands of lines of code. We identify four key challenges: ❶ capturing relevant context, ❷ improving key bug inclusion (KBI), ❸ reducing false alarm rates (FAR), and ❹ integrating human workflows. To tackle these, we propose ❶ code slicing algorithms for context extraction, ❷ a multi-role LLM framework for KBI, ❸ a filtering mechanism for FAR reduction, and ❹ a novel prompt design for better human interaction. Our approach, validated on real-world merge requests from historical fault reports, achieves a 2× improvement over standard LLMs and a 10× gain over previous baselines. While the presented results focus on C++, the underlying framework design leverages language-agnostic principles (e.g., AST-based analysis), suggesting potential for broader applicability.

## 1. Introduction

Code review is essential for improving code quality and detecting defects (Fagan, 2002). Modern Code Review

(MCR) is widely used in open-source (Rigby et al., 2008; 2014; Rigby & Bird, 2013) and industrial settings (Sadowski et al., 2018; Shan et al., 2022), typically involving: (A) code submission, (B) reviewer examination, (C) feedback, and (D) developer revisions.

Despite its benefits, MCR is labor-intensive and time-consuming (Yang et al., 2016), driving research toward automated review comment generation. Existing methods—whether retrieval-based (Gupta & Sundaresan, 2018; Siow et al., 2020; Hong et al., 2022) or deep-learning-driven (Tufano et al., 2021; 2022; Li et al., 2022b;a; Lin et al., 2023; Lu et al., 2023)—often frame it as a **snippet-level code-to-text** task. However, this oversimplification diverges from the core goal of reviewers: detecting defects (Bacchelli & Bird, 2013) (see Section A). Furthermore, current evaluations rely excessively on textual similarity metrics (e.g., BLEU (Papineni et al., 2002), ROUGE (Lin, 2004)), which fail to measure real-world effectiveness (Lu et al., 2025).

**Challenges.** To address these limitations, we investigate a full code review pipeline within a real-world online service (Figure 1). Our system integrates with an internal DevOps platform, generating review reports, filtering comments, and aligning them with code lines. A detailed description of this real-world workflow integration, designed for seamless adoption by developers, is provided in Appendix B. This deployment reveals four key challenges (Appendix C):

*Capturing Proper Code Context:* Effective review requires analyzing dependencies beyond the immediate diff hunk (e.g., variable declarations or method calls). However, excessively long inputs degrade LLM performance, necessitating efficient context extraction.

*Improving Key Bug Inclusion (KBI):* The goal of automated review is to detect critical defects, yet existing methods rely on textual similarity metrics, which fail to measure defect detection capability. More robust evaluation methods, such as Key-Bug Inclusion (KBI), are needed.

*Reducing False Alarm Rates (FAR):* Generative models often produce irrelevant or overly strict comments (e.g., nitpicks, hallucinations), burdening developers. A robust filtering mechanism is required to reduce false positives and enhance signal-to-noise ratio.

*Human-Centric Workflow Integration:* Practical review tools

---

† Work done during the internship or tenure at Kuaishou Technology. [1]Laboratory of Precise Computing, Institute of Software, Chinese Academy of Sciences, Beijing, China [2]University of Chinese Academy of Sciences, Beijing, China [3]Kuaishou Technology, Beijing, China [4]Independent Researcher [5]Sinosoft Company Limited, Beijing, China. Correspondence to: Li Yang <yangli2017@iscas.ac.cn>.

*Proceedings of the 42nd International Conference on Machine Learning*, Vancouver, Canada. PMLR 267, 2025. Copyright 2025 by the author(s).

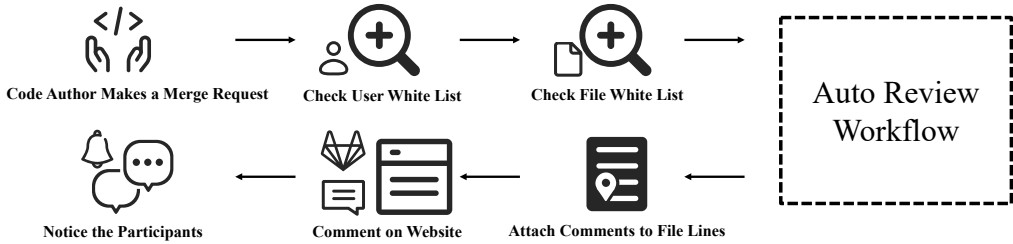

*Figure 1.* The code review automation pipeline integrated into the online service.

must seamlessly integrate into developers' workflows, ensuring comment alignment with code lines while minimizing cognitive overhead. Existing solutions often overlook this critical usability aspect.

**Our Approach.** To address these challenges, we propose: ❶ A static analysis system using code slicing to extract relevant context. ❷ A multi-role LLM framework with chain-of-thought reasoning to enhance defect detection. ❸ A filtering mechanism to eliminate false positive nitpicks and hallucinations. ❹ A line-aware prompt design for precise comment placement.

**Evaluation.** We validate our framework on real-world system failures, including historical core dumps and fault reports that caused significant financial losses. We evaluate it using multiple open-source LLM engines, demonstrating a **2×** performance improvement over standard LLM methods and a **10×** improvement over prior baselines. An ablation study further confirms the contribution of each component, highlighting the impact of code slicing, multi-role reasoning, and filtering mechanisms.

**Contributions.** Our key contributions include being the first to: ❶ **Repository-Level and Merge-Request Granularity**: Elevating automated code review from snippet-level tasks to repository-wide and merge-request (pull-request) granularity. ❷ **Integration with Real-World DevOps Workflows**: Deploying automation into a practical online review system with more practical and objective evaluation metrics beyond text similarity. ❸ **Validation on Industry-Scale Defects**: Demonstrating effectiveness on real-world, high-impact failures in industry-level codebases instead of synthetic test data. ❹ **Code-Review-Oriented LLM Framework**: Designing a specialized framework leveraging code slicing, multi-role collaboration, and filtering mechanisms, achieving substantial improvement in code review performances.

## 2. Background: Code Review Automation

Automating code review is crucial for maintaining software quality by identifying critical bugs early. The goal is to detect severe issues in new merge requests and provide

necessary comments. In 2022, company reports showed that 30% of severe P1+ incidents (asset losses exceeding $350,000) and 24.04% of P4+ incidents stemmed from low-level faults due to inadequate reviews. Even in 2024, change-related core failures accounted for 67% of incidents, with code change-related graded incidents comprising 19.54%, highlighting the urgent need for effective automated review tools. These tools help ensure thorough, compliant reviews, reducing defect risks.

To understand reviewer needs, we surveyed a super reviewer group, summarizing findings in Section D. Background on code slicing and multi-role systems, key techniques in our work, is introduced in Sections E and F.

## 3. Proposed Approach

### 3.1. Overview

Figure 2 illustrates our decoupling process of code review automation architecture: **1) Code Slicing**: Extracting code from the diff hunk within repository context (Section 3.2); **2) Multi-role Code Review System**: Employing a multi-role system to conduct reviews and compile the results (Section 3.3); **3) Redundancy Comment Filter Mechanism**: Filtering out redundant and irrelevant comments to avoid nitpicks and hallucinations (Section 3.4); **4) Line Number Localization**: Ensuring precise identification of code lines where issues occur (Section 3.5). To evaluate the automation, we construct a dataset from historical fault reports, simulating real-world merge requests that introduced defects (Section 3.6).

### 3.2. Code Slicing

Previous work used method-level or diff-level code snippets as independent inputs. However, new code is integrated into a larger codebase during reviews, and understanding the structural context is crucial. We developed a code slicing process that integrates multiple slicing strategies, selectable based on the analysis needs. To avoid redundant slices, we use a caching mechanism to enhance efficiency.

The pseudo code of our slicing algorithms is presented in

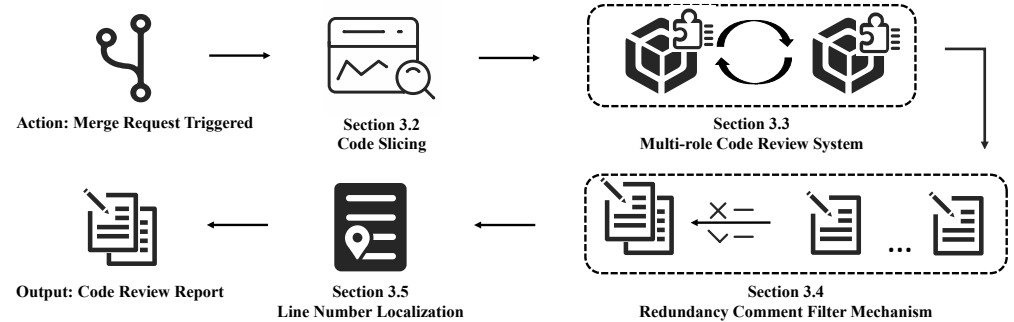

*Figure 2.* An overview of our automated code review workflow.

Section G. Initially, the repository is cloned, and the merge request commit is checked out. A static analysis tool is then applied to generate abstract syntax trees (ASTs), which serve as the foundation for our slicing process. Based on data dependencies and control flow analysis, one or more of the following four optional slicing algorithms may be applied: **1) Original Diff**: The basic code diff without transformations, capturing essential changes in the commit. **2) Parent Function**: Locates the smallest parent function containing the changes, providing functional context. **3) Left Flow**: Tracks the flow of all left-hand values (L-values) in the function and control structures, focusing on the lifecycle of variables. **4) Full Flow**: Extends Left Flow by tracing right-hand values (R-values) and collecting the signatures of callee functions, offering coverage of variable usage and modifications.

### 3.3. Multi-role Code Review System

Our multi-role code review system involves four key roles: Reviewer, Meta-Reviewer, Validator, and Translator. These roles collaborate to enhance the accuracy and efficiency of the review process. The system design is illustrated in Figure 3, and we detail the roles and their processes below.

❶ **Reviewer**: Reviews each code snippet generated by the code slicing algorithm (Section 3.2) and provides detailed comments on potential issues in a predefined format. ❷ **Meta-Reviewer**: Aggregates comments from multiple Reviewers, filtering and sorting them based on predefined thresholds. It merges common issues across reviews. ❸ **Validator**: Validates and refines the merged comments, re-scores them, and ensures that only comments exceeding a certain threshold are retained. ❹ **Translator**: Translates the final comments into the required language for multinational teams, ensuring proper formatting for direct integration into the development environment. Each role is integrated with Chain-of-Thought technique, as detailed in Section H.

### 3.4. Redundancy Comment Filter Mechanism

LLMs often produce an overwhelming number of comments, many of which are either nitpicks or hallucinations. To mitigate this issue, we implemented a Redundancy Comment Filter Mechanism to reduce the number of irrelevant comments.

Our filtering mechanism, integrated within the multi-role system (Section 3.3), operates by answering three key questions for each comment: **Q1: Is this comment a nitpick?** Typical nitpicks include excessive code comments, handling unnecessary edge cases, or overly complex error handling. **Q2: Does the comment identify a fake problem (i.e., a non-existent bug)?** For example, if the comment flags a function call to a known reliable internal library, null pointer checks are considered irrelevant. **Q3: How critical is the issue identified by this comment?** Minor issues, like missing comments, are less severe than potential core dumps or infinite loops.

Each question is rated on a scale from 1 to 7, with 1 indicating a nitpick, fake problem, or minimal issue, and 7 indicating a severe and real issue. The scoring scale (1 to 7) is inspired by other related work (McAleese et al., 2024). We chose this scale to enable a fine-grained and manageable distinction. These scores form the basis of the filtering process throughout the review workflow.

**Coarse Filtering and Sorting by Reviewer.** During the review process, the Reviewer LLMs score each comment based on Q1-Q3. Comments with Q1 or Q2 scores of 4 or below are discarded. This specific threshold was established heuristically to enhance interpretability and has been validated by developer feedback during internal piloting. The remaining comments are then sorted based on their Q3 score and truncated to the Top-N comments.

**Fine Filtering and Sorting by Meta-Reviewer.** The Meta-Reviewer further refines the filtered comments by merging those flagged by multiple Reviewers and removing comments mentioned by only one Reviewer.

*Figure 3.* The multi-role system for automating code review.

**Validation and Re-scoring by Validators.** Validators then re-score the comments by revisiting the original code snippets and applying the same Q1-Q3 criteria. A secondary filter is applied, ensuring that only the most relevant and critical comments proceed to translation and integration into the development platform.

**Integration with the Multi-role System.** The filtered comments are processed by the remaining multi-role components, including translation (if necessary) and final submission to the development platform. This multi-stage process ensures that the delivered comments are both relevant and concise, minimizing redundancy and false alarms. The heuristic approach to threshold definition described herein was chosen to prioritize generalizability, interpretability, and mitigate overfitting in this study. While providing a robust baseline, exploring adaptive or machine-learned thresholds remains a valuable direction for future enhancement to achieve more nuanced filtering.

### 3.5. Line Number Localization

A key challenge overlooked in prior work is the precise localization of comments within the code. Unlike code summarization tasks, code reviews require pinpointing specific lines of code where issues are identified. Without this information, developers face inefficiencies in verifying and addressing comments. For example, the change-involved function has 94.54 lines of code in average based on our statistics, missing line localization can result in significant delays for developers.

We propose a code formatting approach inspired by Aider(Gauthier, 2024), tailored for code review tasks. As shown in Table 1, the format includes an operation label (indicating whether a line is kept, added, or deleted), the line number, and the code content. For non-contiguous code lines, ellipses are used to indicate omissions.

*Table 1.* Code formatting with line position information.

| | |
|---|---|
| **linenumber\|{kept code line}** | Represents lines that remain unchanged. |
| **-linenumber\|{deleted code line}** | Indicates lines that have been removed. |
| **+linenumber\|{added code line}** | Marks newly added lines. |
| **...\|...** | Indicates the omission of non-essential lines. |

### 3.6. Offline Validation

To systematically assess the performance of our system, we developed a dataset curated from the company's fault report platform. Each case in this dataset corresponds to an issue that resulted in actual company losses. For each reported fault, we trace back to the merge request that introduced the fault and its subsequent fixing merge request. Using these, we generate ideal reference comments containing details such as affected files, specific lines of code, fault location, root cause, suggested fix, example code, and issue category. The motivation for conducting such validation is illustrated in Section I.

## 4. Evaluation Design

### 4.1. Research Questions

We define the following research questions (RQs) to guide our evaluation, whose detailed illustrations are in Section J:

*RQ1: How does the overall performance of our framework compare with previous works?*

*RQ2: How do code slicing algorithms impact the performance of the framework?*

*RQ3: How do the different components of the multi-role system impact the performance of our framework?*

*RQ4: How does the redundancy comment filter mechanism address nitpicks and hallucinations?*

*RQ5: How does the representation of line number position information impact overall performance and line number localization success rate?*

### 4.2. Dataset and Studied Models

The primary goal of code review is to prevent problematic code from being merged into the target branch. To simulate real-world code review scenarios, we collected data from a company's core framework team, which is responsible for the production code of the short video recommendation core service. This data was gathered using fault reports recorded on an online platform. These cases come from four repositories and involve total 4,090 developers. By analyzing these reports, we traced the merge requests (MRs) that introduced

the issues and examined the specific commits to reproduce the code snapshots. The detailed statistics are presented in Section K.

Our framework supports multiple LLM engines. To mitigate security risks, we only studied open-source models that can be deployed locally. We exclusively selected large instructed models due to the complex human-instruction-based tasks in our workflow. The final list of models includes: LLaMA-3.1 (70B), Qwen2 (72B), Command R+ (104B), Mistral-large-2407 (123B), and LLaMA3.1 (405B). The reasons for not selecting other models are outlined in Section L.

### 4.3. Metrics

In accordance with the real-world developer expectations discussed in Section D, we evaluate performance at the merge request (MR) level using four metrics, with their formal definitions provided in Section M:

❶ **Key Bug Inclusion (KBI):** Assesses the model's ability to recall critical issues that could lead to tangible losses. ❷ **False Alarm Rate (FAR):** Captures the proportion of irrelevant or erroneous comments, with two variants ($FAR_1$ for all MRs and $FAR_2$ for MRs where key bugs are recalled). ❸ **Comprehensive Performance Index (CPI):** Balances the completeness of key issue detection (KBI) and precision ($100 - FAR$), analogous to the F1-score. It is also computed in two variants ($CPI_1$ and $CPI_2$). ❹ **Line Localization Success Rate (LSR):** Measures the accuracy of line-level references by checking whether comments point to the correct code lines.

### 4.4. Baselines and Experimental Setups

Since our framework focuses on C++, we selected state-of-the-art baselines that support this language: **CodeReviewer** (Li et al., 2022b): A T5 model pre-trained for code review tasks and then fine-tuned. **CCT5** (Lin et al., 2023): A T5 model pre-trained on CodeChangeNet, then fine-tuned. **LLaMA-Reviewer** (Lu et al., 2023): A large LLM fine-tuned for code review tasks based on the LLaMA. **DIS-COREV** (Ben Sghaier & Sahraoui, 2024): A T5 model enhanced via cross-task knowledge distillation for code review. The detailed experimental setups of our framework and baselines are presented in Section N.

## 5. Evaluation Results

### 5.1. RQ1. Comparison with Baselines

We evaluated the performance of our framework on the fault merge request dataset, comparing it with several baseline approaches. Our framework was tested with different large language model (LLM) engines. Our main experiments primarily utilized a homogeneous setup, employing the same

Table 2. Overall performance comparison of our framework using different LLM engines and baseline models. LLM engines marked with $^*$ are quantized. "Val" indicates if Validator role was used.

| Model | Val | KBI↑ | FAR$_1$↓ | CPI$_1$↑ | FAR$_2$↓ | CPI$_2$↑ |
|---|---|---|---|---|---|---|
| *Baselines* | | | | | | |
| CodeReviewer | — | 0.00 | 97.78 | 0.00 | – | – |
| CCT5 | — | 2.22 | 97.58 | 2.32 | 90.91 | 3.57 |
| LLaMA-Reviewer | — | 2.22 | 97.62 | 2.30 | 92.86 | 3.39 |
| DISCOREV | — | 0.00 | 97.78 | 0.00 | – | – |
| *Ours (Left Flow)* | | | | | | |
| LLaMA3.1 | w/o | 20.00 | 84.42 | 17.52 | 66.54 | 25.03 |
| (70B) | w | 2.22 | 37.04 | 4.29 | 66.67 | 4.17 |
| Qwen2 | w/o | 40.00 | 91.21 | 14.42 | 83.57 | 23.29 |
| (72B) | w | 26.67 | 90.63 | 13.87 | 81.52 | 21.83 |
| Command R+ | w/o | 20.00 | 92.99 | 10.38 | 76.08 | 21.78 |
| (103B) | w | 4.44 | 85.00 | 6.86 | 62.50 | 7.95 |
| Mistral-2407 | w/o | 26.67 | 91.07 | 13.38 | 74.85 | 25.89 |
| (123B) | w | 26.67 | 87.07 | 17.41 | 68.18 | 29.01 |
| LLaMA3.1 | w/o | 31.11 | 87.81 | 17.51 | 67.98 | 31.56 |
| (405B)$^*$ | w | 20.00 | 75.37 | 22.07 | 43.52 | 29.54 |
| *Ours (Full Flow)* | | | | | | |
| LLaMA3.1 | w/o | 13.33 | 88.22 | 12.51 | 61.67 | 19.78 |
| (70B) | w | 0.00 | 46.67 | 0.00 | – | – |
| Qwen2 | w/o | 42.22 | 90.99 | 14.86 | 83.91 | 23.30 |
| (72B) | w | 28.89 | 91.73 | 12.86 | 79.06 | 24.28 |
| Command R+ | w/o | 15.56 | 94.22 | 8.43 | 77.14 | 18.51 |
| (103B) | w | 8.89 | 76.30 | 12.93 | 58.33 | 14.65 |
| Mistral-2407 | w/o | 28.89 | 90.68 | 14.09 | 75.44 | 26.55 |
| (123B) | w | 28.89 | 86.11 | 18.76 | 67.30 | 30.68 |
| LLaMA3.1 | w/o | 31.11 | 89.41 | 15.80 | 73.10 | 28.86 |
| (405B)$^*$ | w | 20.00 | 77.96 | 20.97 | 67.59 | 24.73 |

LLM across all roles. This approach was chosen to isolate and clearly assess whether a single, powerful model could effectively address key challenges in code review. Recognizing the practical importance and potential benefits of diverse model deployments, we also conducted extended comparison experiments with heterogeneous LLM assignments for reviewer and validator roles. These experiments, detailed in Appendix O, show that strategic combinations, such as pairing a strong validator with a smaller reviewer, can achieve comparable or even superior performance while potentially optimizing resource usage.

For baselines, since they do not prioritize comments, we evaluated their comments based on whether they passed their respective "quality estimation" filters, which assess whether a code snippet requires a comment. The results are in Table 2.

The results indicate that our framework significantly outperforms the baselines by a factor of 10x across most key metrics, such as key bug inclusion (KBI) and comprehensive performance index (CPI). This marked improvement is likely due to our framework's end-to-end approach to code review automation, which addresses the key challenges of the task and introduces strategies specifically designed to tackle each challenge.

Among the LLM engines tested in our primary setup, LLaMA3.1-405B demonstrated the best overall performance, which aligns with the general scaling laws of language models where capability often increases with parameter count on complex tasks such as code review. However,

our evaluations (detailed in Table 2) also included more compact LLMs. These results show that certain smaller models, particularly those with strong inherent reasoning capabilities, can still achieve competitive performance within our framework. This finding is particularly relevant given the industry trend towards increasing 'capacity density' in newer architectures, where smaller models are progressively narrowing the performance gap. While the largest models may provide peak effectiveness, these observations suggest that a range of LLMs can be effectively utilized, allowing for a balance between performance and computational resource demands, a point further explored in our heterogeneous model assignments (Appendix O).

**Summary of RQ1.** Our framework surpasses baseline approaches significantly (up to 10x on KBI/CPI), thanks to its end-to-end design. LLaMA3.1-405B stands out among tested engines, highlighting the role of model capability. Investigations into heterogeneous LLM combinations also suggest the potential for optimized deployments. (See Appendix T.1 for the extended conclusion.)

### 5.2. RQ2. Effectiveness of Code Slicing

We tested the four code slicing algorithms described in Section 3.2: Original Diff, Parent Function, Left Flow, and Full Flow. It is important to clarify that while our framework does not employ an explicit Retrieval-Augmented Generation (RAG) pipeline, our code slicing mechanism is designed with a RAG-aligned objective. Specifically, it serves a similar purpose to RAG by strategically retrieving and providing the LLM with only the most relevant contextual code 'slices' from the broader codebase. This process aims to focus the model on pertinent information, thereby enhancing its reasoning and effectiveness in the code review task. Our focus in this section is on $KBI$ and $CPI_1$, as these metrics indicate how input content affects the maximum recall capability of LLMs for code review.

The experiments were structured to evaluate the comments generated by the large language models under different conditions, including all comments, comments after applying a coarse filter, and top-k ranked comments (based on scores from Q3). We also tested multi-reviewer settings, where the meta-reviewer merges the comments, and validator settings, where validators further refine the comments. The average results are shown in Table 3, based on the LLaMA3.1-405B-AWQ-Int4 LLM engine. To provide further insight into the variability of these results, the minimum and maximum values for each reported metric across the three runs are detailed in Appendix R.

The results reveal that using only the diff or parent function is less effective, while more detailed slicing (Left Flow and Full Flow) improves performance, especially in key bug inclusion. Surprisingly, Left Flow performs better than

Full Flow, likely due to the large language model's reduced capability when provided with longer contexts, which can cause distraction. This finding supports our assumption that providing targeted and relevant code context is critical for maximizing LLM performance in code review tasks, an observation consistent with the principles underpinning RAG systems where curated information significantly enhances model outputs.

During our analysis of the recalled merge requests (MRs), we found another interesting pattern. Although some slicing algorithms perform worse overall, each algorithm uniquely succeeds in specific cases. This means that each slicing strategy provides valuable context in certain situations. Figure 4 presents a Venn diagram showing the union and differences among the key bugs recalled by each slicing algorithm under the "All" and "+Meta Reviewer" settings. Notably, Left Flow and Full Flow recall most, with significant overlap, but almost each method also uniquely recalls some.

This phenomenon mirrors how human reviewers operate—expanding their focus to different levels of granularity, such as inspecting parent functions or understanding variable usage in different contexts. Some defects are easier to spot in one context, while others require a different view. Therefore, a combination of various slicing strategies might be a promising direction.

**Summary of RQ2.** Left Flow and Full Flow significantly improve key bug inclusion and overall performance compared to simpler slicing. Left Flow often outperforms Full Flow, possibly because shorter context helps maintain focus. Notably, each slicing approach has exclusive successes, suggesting that combining them could further improve detection. (See Appendix T.2 for the extended conclusion.)

### 5.3. RQ3. Effectiveness of Multi-role System

To better understand the capabilities of our multi-role system, we conduct experiments on: ❶ Leveraging the non-determinism of large language models; ❷ The self-correction capability (validator); ❸ The chain-of-thought (CoT) prompting strategy.

#### 5.3.1. NUMBER OF REVIEWERS

Previous research has shown that the non-determinism of large language models (LLMs) can impact results. Specifically, with a best-of-N sampling approach, smaller LLMs can sometimes match or surpass larger models. Since our framework includes a multi-reviewer scenario, where a meta-reviewer merges comments from multiple reviewers, we conduct experiments to assess whether increasing the number of reviewers improves performance.

The results in Table 4 show that increasing the number of reviewers from one to three improves $KBI$ but also

*Table 3.* Impact comparison of different code slicing algorithms on key bug inclusion ($KBI$) and the comprehensive performance index ($CPI$), based on LLaMA3.1-405B-AWQ-Int4. Experiments for a single reviewer are conducted three times to compute the average. "All" represents all comments generated by the reviewer; "Coarse filter" refers to filtering using Q1 and Q2 scores during generation; "Top-k" denotes truncated comments sorted by Q3 scores; "+Meta Reviewer" and "+Validator" settings are evaluated under Top-5 truncation.

| Code Slicing Algorithms | Single Reviewer | | | | | | | | | | Multi Reviewers | | | |
| --- | --- | --- | --- | --- | --- | --- | --- | --- | --- | --- | --- | --- | --- | --- |
| | All | | Coarse Filter | | Top-10 | | Top-5 | | Top-3 | | + Meta Reviewer | | + Validator | |
| | $KBI\uparrow$ | $CPI_1\uparrow$ | $KBI\uparrow$ | $CPI_1\uparrow$ | $KBI\uparrow$ | $CPI_1\uparrow$ | $KBI\uparrow$ | $CPI_1\uparrow$ | $KBI\uparrow$ | $CPI_1\uparrow$ | $KBI\uparrow$ | $CPI_1\uparrow$ | $KBI\uparrow$ | $CPI_1\uparrow$ |
| Original Diff | 23.70 | 5.71 | 17.04 | 12.70 | 16.30 | 12.75 | 14.81 | 12.48 | 11.11 | 10.90 | 13.33 | 5.24 | 11.11 | 10.46 |
| Parent Function | 31.85 | 5.52 | 22.22 | 15.18 | 17.04 | 13.57 | 14.81 | 13.40 | 8.15 | 9.63 | 20.00 | 11.01 | 11.11 | 10.81 |
| Left Flow | 37.04 | 9.77 | 33.33 | 12.80 | 32.59 | 13.94 | 25.93 | 14.26 | 17.78 | 12.77 | 31.11 | 17.51 | 20.00 | 22.07 |
| Full Flow | 39.26 | 9.67 | 32.59 | 11.95 | 31.85 | 13.18 | 25.93 | 13.90 | 13.33 | 10.23 | 31.11 | 15.80 | 20.00 | 20.97 |

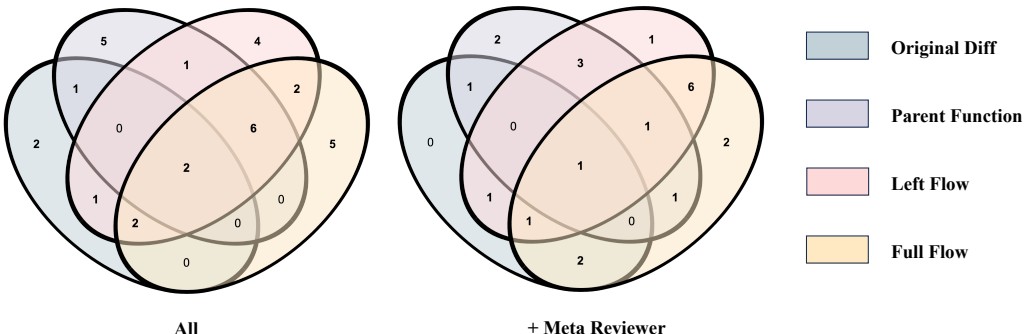

**All**        **+ Meta Reviewer**

*Figure 4.* Venn diagram of recalled key bugs identified by different code slicing algorithms. The "All" setting represents all comments, while the "+Meta Reviewer" setting denotes multi-reviewer comments merged by the meta-reviewer. To analyze per-category performance, a breakdown across logic, security, and performance-related bugs is shown in Appendix P.

*Table 4.* Impact of increasing the number of reviewers from one to three. The "+Meta Reviewer" setting represents the meta-reviewer merging the reviewers' comments, while the "+Validator" setting denotes the validator refining the comments after the meta-reviewer. All settings use Top-5 truncation of reviewer comments.

| Processing Stage | Reviewer Num | KBI↑ | FAR$_1$↓ | CPI$_1$↑ | FAR$_2$↓ | CPI$_2$↑ |
| --- | --- | --- | --- | --- | --- | --- |
| *Original Diff* | | | | | | |
| + Meta Reviewer | 1 | 8.89 | 86.15 | 10.83 | 69.17 | 13.80 |
| | 3 | 13.33 | 96.74 | 5.24 | 75.56 | 17.25 |
| + Validator | 1 | 4.44 | 76.59 | 7.47 | 73.33 | 7.62 |
| | 3 | 11.11 | 90.11 | 10.46 | 71.00 | 16.07 |
| *Parent Function* | | | | | | |
| + Meta Reviewer | 1 | 15.56 | 80.37 | 17.36 | 73.81 | 19.52 |
| | 3 | 20.00 | 92.41 | 11.01 | 73.15 | 22.92 |
| + Validator | 1 | 6.67 | 63.70 | 11.26 | 55.56 | 11.59 |
| | 3 | 11.11 | 89.48 | 10.81 | 65.33 | 16.83 |
| *Left Flow* | | | | | | |
| + Meta Reviewer | 1 | 26.67 | 83.26 | 20.57 | 70.56 | 27.99 |
| | 3 | 31.11 | 87.81 | 17.51 | 67.98 | 31.56 |
| + Validator | 1 | 11.11 | 69.26 | 16.32 | 23.33 | 19.41 |
| | 3 | 20.00 | 75.37 | 22.07 | 43.52 | 29.54 |
| *Full Flow* | | | | | | |
| + Meta Reviewer | 1 | 22.22 | 78.04 | 22.09 | 71.17 | 25.10 |
| | 3 | 31.11 | 89.41 | 15.80 | 73.10 | 28.86 |
| + Validator | 1 | 15.56 | 71.78 | 20.06 | 61.43 | 22.17 |
| | 3 | 20.00 | 77.96 | 20.97 | 67.59 | 24.73 |

leads to higher $FAR_1$ and $FAR_2$, which negatively affect $CPI_1$ and $CPI_2$ in the "+Meta Reviewer" setting. However, after introducing the validator, the performance for three reviewers significantly improves in terms of $CPI_1$ and $CPI_2$. While more reviewers boost $KBI$, they also increase false alarms, making the validator essential to overall performance.

**Summary of RQ3.1.** Increasing the number of reviewers lifts key bug inclusion but raises false alarms. A validator mitigates these alarms, implying a trade-off between coverage and precision. (See Appendix T.3 for extended conclusions.)

### 5.3.2. SELF-CORRECTION ABILITY OF LLMS

In our framework, the validator refines and validates generated comments to correct hallucinations. Table 5 shows that the validator lowers $FAR_1$ and $FAR_2$ but also reduces $KBI$, indicating a trade-off between precision and recall. Our analysis suggests such erroneous rejections of valid comments by validators primarily stem from factors including context propagation from earlier pipeline stages, minor inaccuracies in comment positioning, occasional model input token limits, and inherent scoring variances.

**Summary of RQ3.2.** Self-correction (validator) reduces false alarms but can inadvertently discard critical bug-

*Table 5.* The self-correction ability of LLMs through the Validator role. "w/o" denotes without Validator, "w/" denotes with Validator.

| Validator Status | KBI↑ | $FAR_1$↓ | $CPI_1$↑ | $FAR_2$↓ | $CPI_2$↑ |
|---|---|---|---|---|---|
| *Original Diff* | | | | | |
| w/o | 13.33 | 96.74 | 5.24 | 75.56 | 17.25 |
| w/ | 11.11 | 90.11 | 10.46 | 71.00 | 16.07 |
| *Parent Function* | | | | | |
| w/o | 20.00 | 92.41 | 11.01 | 73.15 | 22.92 |
| w/ | 11.11 | 89.48 | 10.81 | 65.33 | 16.83 |
| *Left Flow* | | | | | |
| w/o | 31.11 | 87.81 | 17.51 | 67.98 | 31.56 |
| w/ | 20.00 | 75.37 | 22.07 | 43.52 | 29.54 |
| *Full Flow* | | | | | |
| w/o | 31.11 | 89.41 | 15.80 | 73.10 | 28.86 |
| w/ | 20.00 | 77.96 | 20.97 | 67.59 | 24.73 |

*Table 6.* Impact of Chain-of-Thought (CoT) on the framework, presenting paired slicing algorithm comparisons. "SR" denotes Single Reviewer, "MR" denotes Multi Reviewers. All multi-reviewer settings use three reviewers and Top-5 truncation.

| Stage | CoT | KBI↑ | $FAR_1$↓ | $CPI_1$↑ | KBI↑ | $FAR_1$↓ | $CPI_1$↑ |
|---|---|---|---|---|---|---|---|
| | | | *Original Diff* | | | *Parent Function* | |
| SR - All | w/o | 20.74 | 97.08 | 5.12 | 31.11 | 96.91 | 5.60 |
| | w/ | 23.70 | 96.74 | 5.71 | 31.85 | 96.98 | 5.52 |
| SR - Top-5 | w/o | 17.04 | 89.17 | 13.05 | 16.30 | 94.57 | 8.03 |
| | w/ | 14.81 | 89.11 | 12.48 | 14.81 | 87.67 | 13.40 |
| MR - Meta | w/o | 15.56 | 82.74 | 16.36 | 17.78 | 89.09 | 13.52 |
| | w/ | 13.33 | 96.74 | 5.24 | 20.00 | 92.41 | 11.01 |
| MR - Val | w/o | 6.67 | 63.89 | 11.26 | 11.11 | 75.19 | 15.35 |
| | w/ | 11.11 | 90.11 | 10.46 | 11.11 | 89.48 | 10.81 |
| | | | *Left Flow* | | | *Full Flow* | |
| SR - All | w/o | 34.81 | 95.16 | 8.49 | 40.00 | 94.40 | 9.83 |
| | w/ | 37.04 | 94.36 | 9.77 | 39.26 | 94.48 | 9.67 |
| SR - Top-5 | w/o | 20.74 | 92.37 | 11.10 | 20.74 | 91.94 | 11.50 |
| | w/ | 25.93 | 90.15 | 14.26 | 25.93 | 90.43 | 13.90 |
| MR - Meta | w/o | 17.78 | 88.74 | 13.79 | 26.67 | 82.81 | 20.90 |
| | w/ | 31.11 | 87.81 | 17.51 | 31.11 | 89.41 | 15.80 |
| MR - Val | w/o | 11.11 | 75.56 | 15.28 | 6.67 | 68.33 | 11.01 |
| | w/ | 20.00 | 75.37 | 22.07 | 20.00 | 77.96 | 20.97 |

detecting comments. Balancing these factors is crucial. (See Appendix T.3 for extended conclusions.)

### 5.3.3. EFFECTIVENESS OF CHAIN-OF-THOUGHT

We compared our specified CoT approach with free-form reasoning. Table 6 shows that CoT prompts often excel in complex slicing tasks (Left Flow, Full Flow), but in simpler tasks (Original Diff, Parent Function), free-form can be just as good or better.

**Summary of RQ3.3.** CoT prompting is especially beneficial in complex contexts. For simpler code slices, the model may perform well without explicit CoT guidance. As more powerful reasoning models, such as GPT-O1 and DeepSeek-R1, emerge, the advantage of specified CoT over free-form reasoning may further diminish. (Appendix T.3)

### 5.4. RQ4. Effectiveness of Comment Filter Mechanism

The comment filter mechanism includes ❶ Coarse reviewer filter, ❷ Top-k truncation, ❸ Meta-reviewer filter, and ❹ Validator validation. Table 7 shows that in flow-based slic-

*Table 7.* The $KBI$, $FAR_1$, and $CPI_1$ results for different code slicing algorithms utilizing our filtering mechanism. This table illustrates the impact of sequential filter stages, including different Top-k truncation values (k=10, 5, 3) for single-reviewer paths. For the multi-reviewer path results shown here (+Meta Reviewer, +Validator), Top-k is set to 5. A comprehensive discussion of Top-k sensitivity, covering both single-reviewer variations and multi-reviewer settings, is presented in Appendix S.

| Reviewer | Filter / Trunc | KBI↑ | $FAR_1$↓ | $CPI_1$↑ | KBI↑ | $FAR_1$↓ | $CPI_1$↑ |
|---|---|---|---|---|---|---|---|
| | | | *Original Diff* | | | *Parent Function* | |
| Single | All | 23.70 | 96.74 | 5.71 | 31.85 | 96.98 | 5.52 |
| | Coarse Filter | 17.04 | 89.72 | 12.70 | 22.22 | 88.17 | 15.18 |
| | Top-10 | 16.30 | 89.43 | 12.75 | 17.04 | 88.53 | 13.57 |
| | Top-5 | 14.81 | 89.11 | 12.48 | 14.81 | 87.67 | 13.40 |
| | Top-3 | 11.11 | 89.14 | 10.90 | 8.15 | 88.15 | 9.63 |
| Multi | + Meta | 13.33 | 96.74 | 5.24 | 20.00 | 92.41 | 11.01 |
| | + Validator | 11.11 | 90.11 | 10.46 | 11.11 | 89.48 | 10.81 |
| | | | *Left Flow* | | | *Full Flow* | |
| Single | All | 37.04 | 94.36 | 9.77 | 39.26 | 94.48 | 9.67 |
| | Coarse Filter | 33.33 | 92.03 | 12.80 | 32.59 | 92.65 | 11.95 |
| | Top-10 | 32.59 | 91.13 | 13.94 | 31.85 | 91.68 | 13.18 |
| | Top-5 | 25.93 | 90.15 | 14.26 | 25.93 | 90.43 | 13.90 |
| | Top-3 | 17.78 | 90.00 | 12.77 | 13.33 | 91.60 | 10.23 |
| Multi | + Meta | 31.11 | 87.81 | 17.51 | 31.11 | 89.41 | 15.80 |
| | + Validator | 20.00 | 75.37 | 22.07 | 20.00 | 77.96 | 20.97 |

ing (Left Flow, Full Flow), adding these filters sequentially decreases $FAR_1$ and improves $CPI_1$. In simpler slicing (Original Diff, Parent Function), only the coarse filter proves particularly effective, likely due to limited context causing more hallucinations. A comprehensive sensitivity analysis of the Top-k truncation hyperparameter $k$—detailing its impact on single-reviewer paths with various $k$ values (as presented in Table 7) and an extended analysis within our multi-reviewer framework—is provided in Appendix S.

**Summary of RQ4.** Our comment filter significantly reduces false alarms and improves performance in more detailed slicing methods. In simpler slicing, the coarse filter stage is the most impactful step. (See Appendix T.4 for the extended conclusion.)

### 5.5. RQ5. Line Number Position

Line number localization is crucial for real-world applications. We tested three formats: **No:** No line position information is provided; **Relative:** Code is provided with a separate list containing relative line positions; and **Inline:** Position information is integrated directly into the code using the format in Table 1.

Table 8 shows that providing line number information (especially inline) significantly improves performance and localization success rate (LSR).

**Summary of RQ5.** Embedding line numbers inline yields the highest performance and LSR, likely because it helps the model anchor comments to specific lines accurately. (See Appendix T.5 for the extended conclusion.)

*Table 8.* Impact of line number position information. "All" represents the average of all comments generated by reviewers, while "+Meta Reviewer" denotes the multi-reviewer workflow with three reviewers and Top-5 truncation. LSR (Line Success Rate) measures whether LLMs provide valid lines, regardless of correctness.

| Position | KBI↑ | FAR$_1$↓ | CPI$_1$↑ | FAR$_2$↓ | CPI$_2$↑ | LSR↑ |
|---|---|---|---|---|---|---|
| *"All" Setting* | | | | | | |
| **No** | 30.37 | 95.66 | 7.58 | 93.12 | 11.17 | 90.54 |
| **Relative** | 42.96 | 94.60 | 9.58 | 92.75 | 12.32 | 92.69 |
| **Inline** | 37.04 | 94.36 | 9.77 | 90.66 | 14.79 | 91.11 |
| *"+ Meta Reviewer" Setting* | | | | | | |
| **No** | 17.78 | 90.70 | 12.21 | 72.71 | 21.53 | – |
| **Relative** | 17.78 | 93.52 | 9.50 | 76.04 | 20.41 | – |
| **Inline** | 31.11 | 87.81 | 17.51 | 67.98 | 31.56 | – |

## 6. Related Work

Code review comments play a crucial role in maintaining software quality, leading to significant research efforts in automating this process. Early studies, such as Gupta & Sundaresan (2018), employed retrieval-based methods, utilizing LSTM models to match new code snippets with historical changes to recommend comments. Siow et al. (2020) advanced this approach by incorporating attention mechanisms to capture semantic nuances more effectively.

With the advent of deep learning, the focus shifted towards automated comment generation. Pioneering efforts by Tufano et al. (2021; 2022) introduced models trained on diverse datasets, including technical texts and code snippets. Subsequent innovations included specialized models such as CodeReviewer (Li et al., 2022b), which leveraged pre-training on code review data, and AUGER (Li et al., 2022a), which used review tags to streamline the task. Another approach, CommentFinder (Hong et al., 2022), presented an efficient retrieval-based model tailored to new code. More recently, LLaMA-Reviewer (Lu et al., 2023) trained large language models specifically for code review tasks, and DISCOREV (Ben Sghaier & Sahraoui, 2024) improved performance by applying cross-task knowledge distillation across successive tasks, and Yu et al. (2024b) focused on fine-tuning LLMs to improve both the accuracy and comprehensibility of automated code reviews. Alongside these advancements in direct comment generation, recent studies have also explored the application of LLMs to other related aspects of the software development lifecycle, such as enhancing code reviewer recommendation (Wang et al., 2024) and automating commit message generation (Tao et al., 2024), underscoring the expanding utility of large models in diverse software engineering contexts.

Despite these advances, previous works have oversimplified the code review process by treating it as a set of snippet-level code-comment pairs. These approaches typically split merge requests into independent snippets and framed the task as a one-to-one neural machine translation (NMT) problem, converting code into natural language. While innova-

tive, this approach provides a limited and idealized view of code review, often evaluated with text similarity metrics, such as BLEU or ROUGE, which do not fully capture the expectations of real-world developers for finding defects.

In practice, code review is more complex, evaluated at the level of entire merge requests of repository codebases rather than individual code-comment pairs. The focus on text similarity fails to consider the broader context, including how comments address the full scope of changes in a MR. Although contributing valuable insights, these studies fall short of replicating the holistic, real-world workflow.

## 7. Conclusion

Motivated by the limitations of prior research that oversimplified code review automation and fell short of practical applications, we explored the complete automation pipeline within a real-world company. We identified and addressed key challenges such as capturing relevant code context, improving key bug inclusion (KBI), reducing false alarm rates (FAR), and integrating human-centric workflows. Our approach introduces four code slicing algorithms, a multi-role LLM framework, a comment filtering mechanism, and a prompt format with inline line number localization. Evaluations on real-world data demonstrated that we significantly outperforms existing methods, achieving up to a 10x improvement in the comprehensive performance index (CPI) over previous baselines.

Key insights include: ❶ Flow-based slicing (Left Flow and Full Flow) provided better context and outperformed simpler methods. ❷ Increasing the number of reviewers improved KBI but required validation to manage false alarms effectively. ❸ The validator role reduced hallucinations but slightly lowered KBI, highlighting a trade-off between precision and recall. ❹ Chain-of-thought guidance proved more valuable in complex slicing scenarios. ❺ Inline line number localization enhanced both comment accuracy and localization success rates.

Looking ahead, four key areas for future research are: ❶ Enhancing code slicing algorithms to capture more relevant context, potentially combining different slicing levels. ❷ Refining LLM interactions and enhancing engine LLM capability to improve key bug recall. ❸ Further optimizing the filtering mechanism, including the investigation of adaptive or learned thresholds, to reduce nitpicks and hallucinations more effectively. ❹ Streamlining pipeline to make automation more accessible.

**Limitation.** We discuss limitations in Section V.

**Data Availability.** We publicly release our codes at `https://zenodo.org/records/14779175`. Details regarding their open-source status can be found in Section U.

## Acknowledgments

This work was supported by the National Key Research and Development Program of China (No. 2023YFB3307202) and the Alliance of International Science Organizations Collaborative Research Program (No. ANSO-CR-KP-2022-03).

## Impact Statement

This work advances the practical defect-focused applications of automated code review. We redefine the code review task by shifting from a snippet-level code-to-text formulation to an end-to-end, merge-request-level codebase analysis. Our approach fills previously overlooked sub-tasks in the automation pipeline and introduces more practical and objective metrics, better aligning the process with developers' expectations for defect detection in real-world software development. These refinements not only establish a more comprehensive foundation for future research in automated code review but also offer insights applicable to other software engineering tasks. Furthermore, our generic framework, including the proposed context slicing algorithms, provides a versatile methodology that can inspire broader applications in code intelligence.

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

## A. The Central Role of Defect Detection in Code Review

Identifying defects has consistently been recognized as the core and most fundamental goal of code review. This understanding aligns with its historical origins, current industry expectations, state-of-the-art research directions, and pressing real-world needs:

- **Historical Foundations:** Defect detection was the original purpose of code reviews, tracing back to the concept of code inspection proposed by Michael Fagan at IBM in 1976 (Fagan, 2002). Fagan Inspections introduced a structured process aimed at reducing long-term costs by detecting and fixing defects early. Subsequent decades of research continued to center on discovering and resolving faults (Ackerman et al., 1989; Sommerville, 2011; Votta Jr, 1993).

- **Contemporary Expectations:** Empirical studies show that both developers and managers consider defect detection the primary expectation of code reviews (Bacchelli & Bird, 2013). For example, a comprehensive survey involving 165 managers and 873 programmers at Microsoft revealed that while code review can serve multiple functions, identifying defects remains the foremost motivation for all stakeholders.

- **Current Research Directions:** Recent state-of-the-art (SOTA) work in automated code review—particularly generation-based methods—continues to emphasize defect detection as the core research objective (Tufano et al., 2024). While these studies simulate code review by injecting known defects, our approach leverages actual, historically documented defects, providing a more realistic and robust evaluation scenario.

- **Real-World Industrial Needs:** Industry practitioners, especially "super reviewers" overseeing thousands of developers and large codebases, highlight an urgent and practical need for more effective defect detection. These expectations were extracted from the Objectives and Key Results set by approximately 50 experienced super reviewers (the concept comes from Kononenko et al.(Kononenko et al., 2016)) within a major production environment of 4000+ developers.

In conclusion, defect detection stands as the central and most essential aspect of code review. While other dimensions—such as code improvement, comprehension, and communication—do appear frequently, they often do not align with the urgent expectations of practitioners. This disconnect between what teams urgently need (effective defect detection) and what reviews often deliver (broader but less critical commentary) warrants a focused research effort (Bacchelli & Bird, 2013).

## B. Detailed Real-World Workflow Integration

As briefly mentioned in the Introduction and illustrated in Figure 1, our automated code review framework is deeply integrated into the real-world Continuous Integration/Continuous Deployment (CI/CD) pipeline and DevOps platform of a large-scale online service company. This appendix provides a more detailed description of this integration, designed to be seamless for developers and provide actionable feedback directly within their existing workflows.

The primary goal of this integration is to automate aspects of the code review process without disrupting established development practices, thereby enhancing both efficiency and code quality. The workflow is triggered upon the submission or update of a Merge Request (MR) within the company's internal DevOps system and proceeds through several automated stages:

1. **MR Trigger and Initial Verification:** When a developer submits an MR, a webhook notifies our automated review system. The system first performs essential verification checks. This includes confirming the submitting user's permissions and ensuring that the changed files fall within the scope of automated review (e.g., correct programming language, project-specific configurations for review). This step is crucial for security, access control, and efficient resource allocation.

2. **Code Analysis and Comment Generation Launch:** Once the MR is verified, the system retrieves the relevant code changes. The core analysis process is then launched:
   - **Code Slicing:** The modified code segments are processed by our code slicing algorithms (detailed in Section 3.2) to extract relevant contextual information necessary for effective review.
   - **Multi-role Review:** These code slices, along with their context, are distributed to our multi-role LLM framework (described in Section 3.3). Each role, potentially with specialized roles (e.g., Reviewer, Validator), analyzes the code to identify potential issues and generate draft review comments.

3. **Comment Filtering and Refinement:** The raw comments generated by the LLM roles undergo a rigorous filtering process using our Redundancy Comment Filter Mechanism (explained in Section 3.4). This multi-stage process (involving Q1-Q3 scoring, coarse filtering, meta-reviewer processing, and validator re-scoring) aims to eliminate nitpicks, false positives, and less critical suggestions, ensuring that only high-quality, actionable comments proceed.

4. **Seamless Injection into DevOps Platform and Developer Notification:** This stage is critical for effective real-world integration:

   - **DevOps System Integration:** The filtered and validated comments are programmatically injected into the company's internal DevOps platform using its provided APIs. Each comment is associated with the specific MR and the relevant commit.
   - **Line-Aware Comment Positioning:** A key feature for developer adoption is the precise positioning of each comment directly at the relevant line number(s) in the diff view of the MR. This is achieved by accurately parsing diff hunks and mapping comment locations. The effectiveness and importance of this line-aware comment injection for providing clear, contextualized, and actionable feedback was specifically evaluated in Section 5.5 (RQ5) and found to be highly valued by developers.
   - **Developer Notification:** Developers (typically the MR author and assigned human reviewers) are notified of the automated review comments through the DevOps platform's standard notification mechanisms (e.g., email, internal messaging/chat system integration). The comments appear within the MR's discussion or review interface, similar to comments made by human colleagues.

This end-to-end automated workflow, from MR submission to the delivery of precisely positioned, filtered comments directly within the developers' familiar DevOps environment, constitutes a seamless integration into their daily activities. It minimizes context switching, presents feedback in an actionable format, and leverages existing platform features for discussion and resolution of the identified issues. The positive developer feedback regarding the non-intrusiveness and utility of the system, particularly due to reliable line-aware comment placement, substantiates our claim of successful real-world workflow integration. This integration was a prerequisite for addressing the key challenges outlined in Section C within a practical, industrial setting.

## C. Four Challenges Identified in Code Review Process

**Challenge 1: Capturing Proper Code Context.** During the merge request process, code is integrated into the repository. Prior studies often split the input into method-level or diff hunk-level segments, which introduces two major problems: ❶ the omission of critical context beyond the method, hunk, or even file level, such as variable declarations, definitions, and assignments—particularly in languages like C++—which misleads large language models (LLMs) into generating false alarms or missing key bugs; and ❷ the truncation of long code snippets, leading to significant omissions. However, feeding excessively long inputs to LLMs also degrades performance due to the models' sensitivity to the relative position of the target sequence (Hsieh et al., 2024; Liu et al., 2024). Therefore, finding an optimal way to capture proper code context is essential.

**Challenge 2: Improving Key Bug Inclusion (KBI).** A primary goal of automated code review is to identify key bugs introduced by new merge requests that could compromise system reliability and performance. In 2022, 30% of severe P1+ incidents (asset loss > $350,000) and 24.04% of P4+ incidents in the company were attributed to preventable low-level faults, highlighting the need for robust code reviews and automated systems. Prior studies have validated models based on overall text similarity metrics, which are often misleading. Text similarity does not necessarily correlate with a model's recall ability, as higher similarity scores may reflect linguistic style rather than the identification of critical bugs(Lu et al., 2025). OpenAI researchers have proposed Critique-Bug Inclusion (CBI) as a key metric for evaluating LLM performance (McAleese et al., 2024), which we adopt for our task. Increasing the KBI capability of the framework is a key focus of our approach.

**Challenge 3: Reducing False Alarm Rates (FAR).** Generative models frequently produce redundant or irrelevant comments, including nitpicks and hallucinations, even for a single code snippet (McAleese et al., 2024). In real-world scenarios, merge requests often contain numerous code snippets, and managing redundant comments can overwhelm human reviewers. Most previous studies (Gupta & Sundaresan, 2018; Siow et al., 2020; Tufano et al., 2021; 2022; Li et al., 2022a; Hong et al., 2022) have not addressed this issue, simplifying the review task to merely generating natural language comments from problematic code. Some works attempted to introduce discriminators to assess comment quality (Lin et al., 2023; Li et al., 2022b), but

these approaches have been shown to be ineffective (McAleese et al., 2024). Thus, a robust filtering mechanism is needed to minimize redundant comments and reduce false alarms.

**Challenge 4: Human-Centric Workflow Integration.** Simplified code review tasks often overlook essential real-world workflow components, such as attaching comments to specific code lines. This step reduces the cognitive load on developers by making it easier to verify the validity of comments. Despite its importance, this aspect is frequently ignored in previous work (Gupta & Sundaresan, 2018; Siow et al., 2020; Tufano et al., 2021; 2022; Li et al., 2022a; Hong et al., 2022; Lin et al., 2023; Li et al., 2022b), which hinders real-world usability. Incorporating such functionality is critical for improving user experience alongside KBI and FAR.

## D. Demands and Expectations from Reviewers

We conducted qualitative studies, including surveys and in-depth interviews, to understand the expectations and demands for automated code review from a group of 50 experienced "super reviewers" within a large technology company's core infrastructure teams. From the reviewers' perspective, an effective automation solution must fulfill two primary demands regarding its operational behavior and meet three key expectations concerning the quality and utility of its feedback:

The two demands, which aim to simulate desirable aspects of human review, are:

- **D1:** Automation should operate at the merge request level, reviewing all changes holistically rather than focusing on partial code snippets or individual diff hunks.

- **D2:** Automation should incorporate global repository knowledge beyond the diff, enabling access to broader information such as variable declarations or function definitions not included in the immediate changes.

The three key expectations for the generated feedback are:

- **E1:** Identify as many key issues as possible, thereby reducing the number of missed critical bugs. This directly informed our Key Bug Inclusion (KBI) metric, introduced in the Introduction. In our interviews, developers consistently emphasized that "catching critical bugs" is the absolute top priority for any automated review tool.

- **E2:** Minimize nitpicking comments and hallucinations (i.e., non-existent issues), which corresponds to reducing the False Alarm Rate (FAR). This is crucial for lessening the burden on human reviewers when verifying automated feedback. Developers particularly stressed this point, noting that reducing irrelevant comments is vital for the adoption and continued trust in an automated system, with one sentiment being that "even one false positive can erode trust." Initial feedback from the system's current deployment within an internal development team has further underscored this, revealing FAR to be an especially sensitive metric directly impacting developer perception and engagement.

- **E3:** Support human-centric interaction by attaching comments precisely to the correct code lines. This aids reviewers or code owners in efficiently verifying issues and prevents confusion or misleading information, a factor captured by our Line Localization Success Rate (LSR).

The demand **D2** aligns with Challenge 1 (Capturing Relevant Context) from the Introduction, while expectations **E1**, **E2**, and **E3** correspond to Challenges 2 (Improving KBI), 3 (Reducing FAR), and 4 (Integrating Human Workflows) respectively. Previous work primarily focused on aspects related to **E1**, often using text similarity as an indirect proxy, but this does not guarantee precision or the detection of key bugs, rendering such approaches less suitable for demanding real-world applications. In contrast, our approach, informed by these direct developer insights, addresses these challenges comprehensively. We introduce a set of merge-request level evaluation metrics tailored to these real-world demands, including Key Bug Inclusion (KBI), False Alarm Rate (FAR), Comprehensive Performance Index (CPI), and Line Localization Success Rate (LSR).

While our metrics and system design are thus grounded in extensive interactions with professional developers and initial deployment feedback, we acknowledge that formal, systematic user studies evaluating perceived review quality and overall system utility have not yet been conducted. Such studies represent an important avenue for future work to further strengthen the validation of our approach.

## E. Background: Code Slicing

Providing the entire repository as input ensures comprehensive context, but large language models (LLMs) have token limitations. Increasing the input size within these constraints can reduce model performance and lead to delays in inference (Hsieh et al., 2024; Liu et al., 2024). Code slicing offers a potential solution by efficiently providing sufficient context while remaining concise. This technique uses static analysis to form code units, parsing code into an abstract syntax tree and slicing it based on node relationships. For large code snippets, code slicing divides them into small, independent segments. For small snippets, like diff hunks in code reviews, it enriches the context by retrieving statements and variable usages related to the changes. While code slicing has been effectively applied to tasks such as vulnerability detection (Li et al., 2018; 2021; Cheng et al., 2021; Yu et al., 2023), its explicit and systematic integration into LLM-based automated code review pipelines—specifically for enhancing context-aware defect localization and guiding comment generation—appears to be less explored in prior literature. While Lu et al. (2024) made initial attempts at testing context components, the bulk of prior research in automated code review has remained largely centered on snippet-level analysis or comment naturalness. Consequently, the richer repository-level contextual understanding, such as that afforded by code slicing, has often been underutilized. To address real-world needs and explore this promising direction, we designed four code slicing algorithms that provide appropriate context for automated reviews.

## F. Background: Large Language Models and Multi-role Systems for Software Engineering

Transformer-based large language models (LLMs)(Vaswani et al., 2017) have achieved notable success in natural language processing tasks(Ray, 2023). The ongoing evolution and refinement of these models involve diverse research efforts, including explorations into advanced knowledge integration techniques (Liang et al., 2025) and the development of robust safeguards against potential misuses (Jiang et al., 2025). Given that LLMs now often include code in their training corpora, they have developed strong abilities in code-related tasks alongside their general language capabilities. This blurs the line between code-specific and general models. For instance, GPT-4 excels in code generation, while models like ChatGPT (Achiam et al., 2023) and LLaMA (Touvron et al., 2023) have shown potential in generating commit messages (Zhang et al., 2024), tests (Chen et al., 2024b; Schäfer et al., 2023), method renaming (AlOmar et al., 2024), log analysis (Ma et al., 2024a;b), and smart contract vulnerability detection(Yu et al., 2024a; 2025; Yuan et al., 2025).

Complex tasks like automated code review, which demand deep-level reasoning, often exceed the capabilities of individual LLMs. Multi-role systems have emerged as an effective approach to decompose and tackle such tasks, assigning specialized roles to different LLM roles that collaborate to solve the overall problem (Guo et al., 2024; Chen et al., 2024a). In this work, we adopt a multi-role system utilizing mainstream open-source LLMs with large parameter counts to enhance Key Bug Inclusion (KBI) and filter redundant comments, thereby making the code review process more efficient and precise.

## G. Code Slicing Algorithms

The code slicing process in this paper is composed of several interrelated steps designed to isolate, identify, and group code statements relevant to a given set of changes. Specifically, the detailed algorithms of our approach are presented in Algorithms 1–9, each focusing on a different aspect or mode of slice generation. These algorithms work together as follows:

- **Algorithm 1 (CodeSlicing)** drives the entire process: It clones the target repository at the specified commit, initializes data structures, and orchestrates the slicing workflow for each file.

- **Algorithm 2 (ProcessAST)** processes each file's Abstract Syntax Tree (AST) using a chosen slicing option, determining which statements intersect with the diff and storing those statements in a cache.

- **Algorithm 3 (GenerateNewSlice)** takes contiguous segments of diff statements as seeds and expands them according to the chosen slicing strategy, removing statements from the cache once they are incorporated into a slice.

- **Algorithm 4 (GetContiguousDiffSegment)** serves as a helper function that extracts a cohesive set of adjacent statements from the cache, ensuring that logically connected changes are processed together.

- **Algorithm 5 (ApplySlicingAlgorithm)** acts as a dispatcher, selecting one of four specific slicing methods based on the chosen option:

  - **Algorithm 6 (OriginalDiff)**: Focuses on the original diff statements and their direct dependencies.

- **Algorithm 7 (ParentFunction)**: Retrieves the smallest function containing all diff statements, providing function-level context for the slice.
- **Algorithm 8 (LeftFlow)**: Performs a backward data-flow trace from the diff statements by analyzing L-values and their defining statements.
- **Algorithm 9 (FullFlow)**: Extends `LeftFlow` by also including forward data-flow tracing for R-values and callees, capturing both backward and forward dependencies.

Through this sequence of algorithms, relevant code segments are iteratively located, expanded, and grouped, resulting in a collection of slices tailored to the user's chosen level of context or detail.

---

**Algorithm 1** CodeSlicing

---

1:  **main** CodeSlicing
2:      **Input:** repository, commit, slicingOption
3:      **Output:** slices
4:      clone repository and checkout to commit
5:      $ASTs \leftarrow$ ApplyStaticAnalysisTool(repository)
6:      $cache \leftarrow$ InitializeCache()
7:      $slices \leftarrow$ [] {Initialize slice list}
8:      **for** each $AST$ in $ASTs$ **do**
9:          ProcessAST($AST$, slicingOption, cache)
10:     **end for**
11:     **while** not cache.isEmpty() **do**
12:         $seed \leftarrow$ GetContiguousDiffSegment($cache$)
13:         $newSlice \leftarrow$ GenerateNewSlice($seed, cache, slicingOption$)
14:         $slices$.append($newSlice$)
15:     **end while**
16: **end main**

---

**Algorithm 2** ProcessAST

---

1:  **function** ProcessAST
2:      **Input:** AST, option, cache
3:      **Output:** updated cache
4:      $slice \leftarrow$ ApplySlicingAlgorithm($AST, option$)
5:      **for** each $statement$ in $slice$ **do**
6:          **if** $statement$ intersects with diff **then**
7:              $cache$.add($statement$)
8:          **end if**
9:      **end for**
10: **end function**

---

## H. Integration of Chain-of-Thought (CoT) in the Review Process

We integrate Chain-of-Thought (CoT) prompts to guide the roles in the review process. Below are the CoT prompts for each:

**For the Reviewer:**

1. **System Introduction:** Introduces the task and provides guidance related to the code repository and input format.

2. **Understand:** Helps the model comprehend the purpose of the code changes.

3. **Analyze:** Instructs the model to analyze the code for defects or performance issues.

4. **Re-evaluate:** Guides the model to review its analysis, minimizing nitpicks and hallucinations. Three specific questions are posed to quantify nitpicks, hallucinations, and severity, inspired by (McAleese et al., 2024).

---

**Algorithm 3** GenerateNewSlice

---
1: **function** GenerateNewSlice
2:    **Input:** seed, cache, option
3:    **Output:** newSlice
4:    $newSlice \leftarrow []$ {Start forming a new slice from the seed}
5:    Add seed statements to $newSlice$
6:    Remove seed statements from $cache$
7:    **for** each $statement$ in $newSlice$ **do**
8:       Expand the slice {Expanding the slice}
9:       $expandedSlice \leftarrow$ ApplySlicingAlgorithm($statement, option$)
10:       **for** each $expStatement$ in $expandedSlice$ **do**
11:          **if** $expStatement$ is in $cache$ **then**
12:             $newSlice$.append($expStatement$)
13:             $cache$.remove($expStatement$)
14:          **end if**
15:       **end for**
16:    **end for**
17: **end function**

---

**Algorithm 4** GetContiguousDiffSegment

---
1: **function** GetContiguousDiffSegment
2:    **Input:** cache
3:    **Output:** contiguousSegment
4:    $contiguousSegment \leftarrow$ Extract a contiguous segment of cached diff statements
5: **end function**

---

**Algorithm 5** ApplySlicingAlgorithm

---
1: **function** ApplySlicingAlgorithm
2:    **Input:** AST, option
3:    **Output:** sliced statements
4:    **switch** (option)
5:       **case** "OriginalDiff":
6:          OriginalDiff(AST)
7:       **case** "ParentFunction":
8:          ParentFunction(AST)
9:       **case** "LeftFlow":
10:          LeftFlow(AST)
11:       **case** "FullFlow":
12:          FullFlow(AST)
13:    **end switch**
14: **end function**

---

**Algorithm 6** OriginalDiff

---
1: **function** OriginalDiff
2:    **Input:** AST
3:    **Output:** sliced set $S$
4:    $D \leftarrow$ diff statements in $AST$
5:    $S \leftarrow \emptyset$
6:    **for** each $d$ in $D$ **do**
7:       $S \leftarrow S \cup \{d\} \cup$ dependencies of $d$
8:    **end for**
9: **end function**

---

---

**Algorithm 7** ParentFunction

---
1: **function** ParentFunction
2:    **Input:** AST
3:    **Output:** sliced set $S$
4:    $D \leftarrow$ diff statements in $AST$
5:    $F \leftarrow$ smallest function containing all $D$
6:    $S \leftarrow$ all statements and declarations in $F$
7: **end function**

---

**Algorithm 8** LeftFlow

---
1: **function** LeftFlow
2:    **Input:** AST
3:    **Output:** sliced set $S$
4:    $D \leftarrow$ diff statements in $AST$
5:    $S \leftarrow \emptyset$
6:    **for** each $d$ in $D$ **do**
7:       $L \leftarrow$ all L-values affected by $d$
8:       **for** each $l$ in $L$ **do**
9:          $S \leftarrow S \cup$ backward trace of $l$
10:       **end for**
11:    **end for**
12: **end function**

---

**Algorithm 9** FullFlow

---
1: **function** FullFlow
2:    **Input:** AST
3:    **Output:** sliced set $S$
4:    $D \leftarrow$ diff statements in $AST$
5:    $S \leftarrow$ LeftFlow($AST$)
6:    **for** each $d$ in $D$ **do**
7:       $R \leftarrow$ all R-values and callees affected by $d$
8:       **for** each $r$ in $R$ **do**
9:          $S \leftarrow S \cup$ forward trace of $r$
10:       **end for**
11:    **end for**
12: **end function**

---

5. **Organize Your Thoughts:** Directs the model to write a detailed review comment specifying the issue, affected lines, root cause, recommended solution, and example code.

6. **Final Comment:** Instructs the model to output the final comment in JSON format.

**For the Meta-Reviewer:**

1. **System Introduction:** Introduces the task of merging Reviewer comments and provides guidelines on the required format.

2. **Analyze:** Instructs the model to analyze Reviewer comments, focusing on patterns, discrepancies, and insights.

3. **Organize and Sort Final Comments:** Guides the model to format the refined comments in a prioritized JSON list, calculating the overall scores and sorting by criticality.

**For the Validator:**

1. **System Introduction:** Similar to the Reviewer, but with a focus on accuracy and relevance.

2. **Validate the Comment:** Guides the model to review and validate the existing comments, aiming to reduce false alarms.

3. **Refine the Comment:** Ensures the comment is refined for clarity and correctness.

4. **Final Comment:** Outputs the validated comment in a JSON format suitable for the development environment.

**For the Translator:**

1. **System Introduction:** Introduces the translation task and explains the input format.

2. **Translation and Formatting Requirements:** Guides the model to translate items into the target language, ensuring proper formatting.

3. **Translated Comments:** Outputs the translated comments in JSON format for direct integration into the development environment.

## I. Rationale for Offline Validation

The primary goal of our review system is to recall as many historical faults as possible while minimizing irrelevant comments that could burden developers. Each recalled fault suggests that our system could potentially prevent similar future issues.

To evaluate the system, we use several key performance metrics, including the key bug inclusion rate (KBI), false alarm rate (FAR), comprehensive performance index (CPI), and line localization success rate (LSR), which are defined in Section 4.3. The CPI serves as an overall measure, balancing the trade-off between effective bug detection (KBI) and minimizing false alarms (FAR).

We hypothesize that this validation approach, which goes beyond simple text similarity metrics, provides a more accurate measure of the system's ability to handle real-world code review challenges. For instance, a change from "and" logic to "or" in code can fundamentally alter the program's behavior, but such a subtle difference may be missed by conventional text-based comparisons. Additionally, we observed that the same issue is often described in multiple ways, complicating simple text-based comparisons and highlighting the need for more nuanced metrics.

## J. Research Questions in Detail

*RQ1: How does the overall performance of our framework compare with previous works?* This question evaluates our framework's overall performance against existing baselines, focusing on metrics such as KBI, FAR, and CPI (defined in Section 4.3). Since our framework decouples from base LLMs, we utilize various open-source large language model

(LLM) engines to provide a comprehensive evaluation. Specifically, for RQ2-5, We conduct ablation studies using the LLaMA3.1-405B LLM engine, which is one of the most representative models.

*RQ2: How do code slicing algorithms impact the performance of the framework?* This question investigates the effect of code slicing on the code review process. As the first to apply code slicing in this context, we compare the effectiveness of four slicing algorithms and study the results using a Venn diagram. This analysis focuses primarily on KBI and CPI, particularly the ability to recall key bugs.

*RQ3: How do the different components of the multi-role system impact the performance of our framework?* This question explores the influence of various components of our multi-role system, including the number of reviewers, the self-correction capability of LLMs (validator), and the impact of Chain-of-Thought (CoT) prompts. We measure KBI, FAR, and CPI to assess overall performance.

*RQ4: How does the redundancy comment filter mechanism address nitpicks and hallucinations?* This question evaluates the effectiveness of our redundancy comment filter mechanism in reducing nitpicks and hallucinations. We sequentially evaluate the contribution of each component of the filter mechanism, with a focus on KBI, FAR, and CPI.

*RQ5: How does the representation of line number position information impact overall performance and line number localization success rate?* This question assesses the impact of our code representation format on overall performance and the line localization success rate (LSR). We compare our method with two other formats: no line position information and supplementary line position in a separate list. Here, LSR is considered alongside KBI, FAR, and CPI.

## K. Dataset Statistics

A key point that may lead to misunderstanding is that our evaluation cases operate at the merge-request level, rather than focusing on isolated code snippets. Over the past three years (starting from 2021), we have collected all recorded faulty merge requests from four repositories maintained by over 4,000 developers. Fault selection follows a practical criterion: each fault must have caused a user-visible issue and been formally logged in the company's internal defect tracking system (Section 3.6). This result-oriented strategy emphasizes real impact, even if it does not fully cover all C++ error types. Each merge request often involves multiple, interdependent code changes, making the evaluation scenario more complex and realistic than snippet-level analyses.

Our focus is on C++ code, as it represents a critical portion of the company's software infrastructure. The dataset consists of 45 real-world fault reports, each corresponding to a significant issue that caused financial losses, along with the associated merge request snapshots. Among these 45 cases, 12 are logic errors, 31 are code security errors, and 2 are performance-related errors. We have released a desensitized JSON folder of fault descriptions in our Zenodo repository.[1] The dataset includes both edge and typical cases, e.g.:

- Case 4694_23117: array out-of-bounds and null-pointer dereference.

- Case 16231_13308: misuse of `boost::random::beta_distribution`.

On average, each merge request includes 8.02 changed C++ files, with 416.8 newly added lines spread across 14.84 modified functions (a total of 1,403.36 lines affected). Unlike previous datasets that focus on individual code snippets, the faults in our dataset span multiple, interconnected code changes at the merge-request level.

The CodeReviewer dataset (Li et al., 2022b) is a well-known benchmark for code review comment generation. However, we do not adopt it because (1) it is a snippet-level dataset lacking repository-level context, and (2) it does not focus on defect detection and lacks structured fault reports for comprehensive understanding (Lu et al., 2025). These limitations reduce its applicability to real-world, defect-focused code review automation.

To provide context, we compare our dataset's scale with that of CodeReviewer. Table 9 juxtaposes the size and complexity of the datasets, demonstrating that ours is of comparable magnitude, but situated at a more realistic granularity (merge-request level) that better reflects practical development workflows.

---

[1] https://zenodo.org/records/14779175

*Table 9.* Dataset statistics. The statistics of the CodeReviewer dataset are estimated based on the language distribution (Li et al., 2022b) and the distribution of comment types (Bacchelli & Bird, 2013).

| Dataset | # of Code Snippets | LOC (Lines of Code) |
| --- | --- | --- |
| CodeReviewer Test (C++ Subset) | 814 | ∼147k |
| CodeReviewer Test (C++ & Defects Subset) | 114 | ∼21k |
| Ours (Merge-Request Level) | 668 | ∼63k |

## L. Excluded Models and Justifications

Our framework is designed to be model-agnostic and independent of any specific base model. However, for the purposes of our study, we only present results on selected representative open-source models. Below, we outline the reasons for not including certain other models in our experiments:

- **Closed-Source Models:** This category includes proprietary models such as the GPT series and the Claude series. These models remain inaccessible for self deployment due to their closed nature, and utilizing fault reports for evaluation poses potential security and data transmission risks.

- **Small and Weak Models:** We also experimented with several smaller or less powerful models as the base model for our framework, including Gemma-2-27B, CodeGemma-7B, MiniCPM3-4B, GLM-4-9B-Chat, CodeGeeX4-All-9B, DeepSeek-V2-Lite-Chat, DeepSeek-Coder-V2-Lite, CodeLlama-70B, Phi-3-Medium-128K, Phi-3.5-MoE, and Aya-23-35B. Unfortunately, these models frequently failed due to their limited capabilities, making it unfair to include them in comparisons under such conditions.

- **Excessively Large Models:** Our experiments were conducted on an infrastructure consisting of eight A100-40G GPUs. However, since FP8 quantization is not supported on Ampere-architecture A100s, we were unable to deploy mixture-of-experts (MoE) models with a very high parameter count. This limitation prevented us from testing models such as DeepSeek-V2-Chat-0628, DeepSeek-Coder-V2, and Mixtral-8x22B-Instruct.

## M. Metric Formulations

We intentionally avoid BLEU and ROUGE due to their limitations in evaluating the quality of code review comments:

1. Our task involves many-to-many mappings between code and reviews, violating BLEU's single-reference assumption.

2. Code review requires reasoning and domain expertise; recent studies show that BLEU and ROUGE fail to reflect quality in such tasks.

3. Real fault reports and LLM-generated comments differ significantly in style and expression, making surface-level textual similarity unreliable.

Regarding vagueness: rather than evaluating linguistic style, we focus on outcome-based metrics that directly reflect the effectiveness of review comments. Specifically, we propose three core metrics—**Key Bug Inclusion (KBI)**, **False Alarm Rate (FAR)**, and the composite **Comprehensive Performance Index (CPI)**—which are objective, interpretable, and domain-relevant. Further discussion on their behavior and limitations is provided in Appendix Q.

### M.1. Key Bug Inclusion (KBI)

By "key bugs," we refer to issues that can lead to tangible losses (e.g., performance degradation or potential future failures), even if the negative impact is not immediate. The P1-P4 incidents mentioned in our dataset (Section 4.2) all qualify as key bugs. This framing ensures our focus remains on high-impact defects rather than trivial concerns. KBI measures the model's ability to recall key issues that cause system faults. It is calculated as the percentage of recalled key issues out of the total issue set:

$$\text{KBI} = \frac{\text{Number of recalled key issues}}{\text{Total number of key issues}} \times 100 \tag{1}$$

## M.2. False Alarm Rate (FAR)

FAR evaluates the extent to which the model generates irrelevant or erroneous comments (false alarms). We consider all comments unrelated to key issues mentioned in the fault reports as false alarms. FAR is calculated as the percentage of false alarm comments relative to total comments. Two types of FAR are defined:

1. $FAR_1$: Calculates the False Alarm Rate for each individual MR and then averages these rates across all MRs to provide an overall measure of the model's ability to avoid false alarms.

$$FAR_1 = \frac{1}{N} \sum_{i=1}^{N} \left( \frac{\text{Number of false alarm comments in MR}_i}{\text{Total number of comments in MR}_i} \times 100 \right) \tag{2}$$

2. $FAR_2$: Focuses on MRs where key bugs were successfully recalled, offering insight into precision when key issues are identified.

$$FAR_2 = \frac{1}{M} \sum_{j=1}^{M} \left( \frac{\text{Number of false alarm comments in recalled MR}_j}{\text{Total number of comments in recalled MR}_j} \times 100 \right) \tag{3}$$

Where $N$ is the total number of MRs, and $M$ is the number of MRs where key bugs were recalled.

## M.3. Comprehensive Performance Index (CPI)

To balance KBI and FAR, we propose the Comprehensive Performance Index (CPI), which harmonizes KBI and FAR in a similar manner to the F1-Score. CPI evaluates the completeness of key issue detection and the precision of the model's comments. Two versions of CPI:

1. $CPI_1$: Based on $FAR_1$, considering all MRs.

$$\text{CPI}_1 = 2 \times \frac{\text{KBI} \times (100 \text{ - } \text{FAR}_1)}{\text{KBI} + (100 \text{ - } \text{FAR}_1)} \tag{4}$$

2. $CPI_2$: Based on $FAR_2$, focusing on MRs where key bugs were recalled.

$$\text{CPI}_2 = 2 \times \frac{\text{KBI} \times (100 \text{ - } \text{FAR}_2)}{\text{KBI} + (100 \text{ - } \text{FAR}_2)} \tag{5}$$

## M.4. Line Localization Success Rate (LSR)

LSR evaluates the model's ability to successfully associate comments with the valid code lines. A success is recorded if the correct line number is provided and valid in the code. LSR is calculated as the percentage of correct line number cases:

$$\text{LSR} = \frac{1}{N} \sum_{i=1}^{N} \left( \frac{\text{Number of correct line number cases in MR}_i}{\text{Total number of comments in MR}_i} \times 100 \right) \tag{6}$$

# N. Experimental Setups in Detail

The code slicing component of our framework is implemented using Cppcheck(Marjamäki, 2024), while the LLM engines are integrated through an API supported by the vLLM framework (Kwon et al., 2023), and baselines are integrated via Flask(Organization, 2024). All models and baselines are hosted on a server equipped with an AMD EPYC 7702 CPU and eight Nvidia A100-40G GPUs. For large models such as LLaMA3.1-405B, we utilize an Int4 version quantized using AWQ (Lin et al., 2024). For other models, we use the original half-precision floating-point format (FP16). *For context on overall throughput for large-scale evaluations, processing our entire dataset, which includes many large merge requests (often over 20 modified files each), with models having 120B activated parameters took approximately 9 hours on our server setup.*

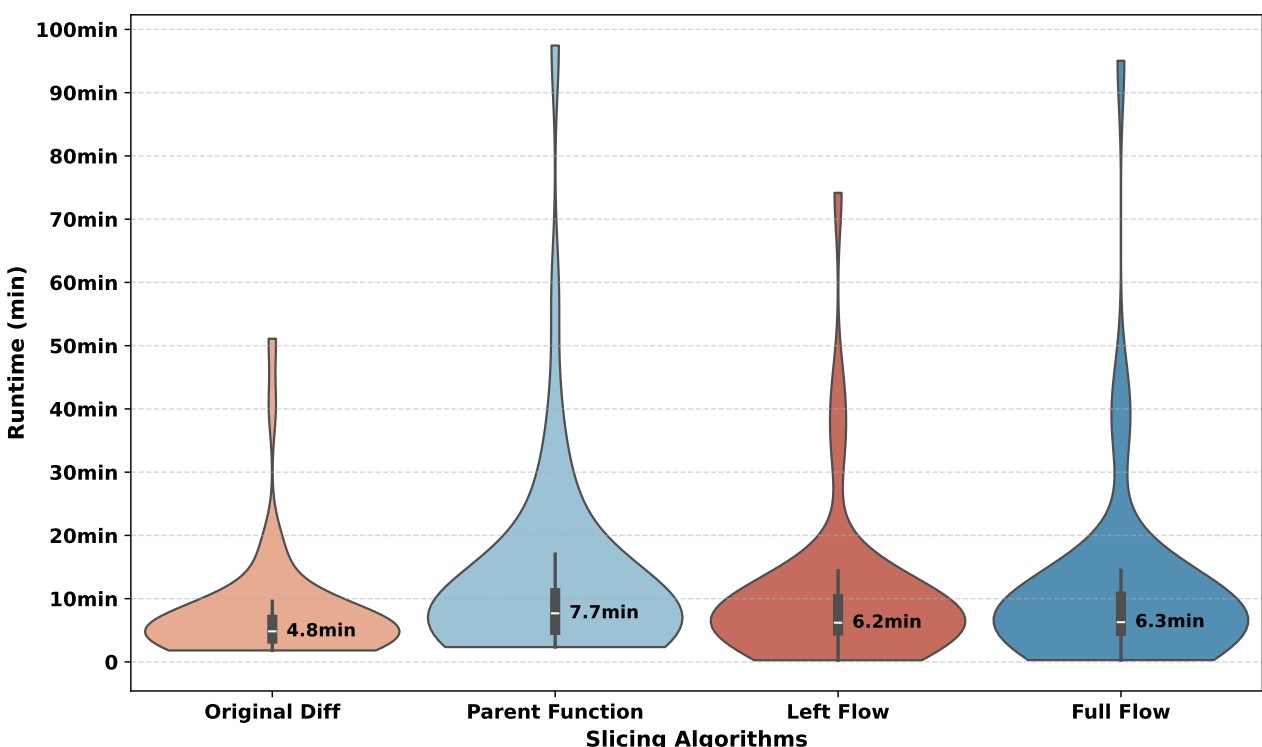

*Figure 5.* Runtime per merge request under different slicing algorithms and LLaMA3.1-405B as base models. The most time-consuming algorithm is *Function*, due to its inclusion of the largest extra context. However, all runtimes are within an acceptable range based on our analysis.

Regarding typical per-merge request runtime, we report detailed timings using violin plots. Figure 5 illustrates these distributions.

The median runtime per MR is 6.2 minutes. The overall CI/CD pipeline (including compilation, analysis, and deployment checks) typically takes 15–30 minutes. Our module runs in parallel from the beginning and does not introduce blocking delays. Thus, we believe the overhead is acceptable in practical scenarios.

## O. Performance of Heterogeneous Model Combinations

While our primary experiments (Section 5.1) employ a homogeneous Large Language Model (LLM) setup across all roles to isolate the impact of a single powerful model, exploring heterogeneous model combinations offers the potential for optimizing both performance and resource utilization. This section details supplementary experiments where LLM assignments for reviewer and validator roles were varied. The results, presented in Table 10, demonstrate that strategically pairing a strong validator model with a computationally less intensive (i.e., smaller) reviewer model can yield performance comparable or, in certain metrics, even superior to configurations relying solely on a powerful model for both roles.

*Table 10.* Performance under different reviewer-validator model combinations and slicing algorithms. The results suggest that the validator plays a more critical role, as it is closer to the final decision output. Interestingly, a combination of a weaker reviewer and a stronger validator can achieve comparable or even superior performance, indicating potential room for improvement through heterogeneous model pairing.

| Slicing Algorithm | KBI↑ | FAR$_1$↓ | CPI$_1$↑ | FAR$_2$↓ | CPI$_2$↑ |
|---|---|---|---|---|---|
| *LLaMA3.1-405B as reviewer, LLaMA3.1-70B as validator* | | | | | |
| Original Diff | 2.22 | 42.22 | 4.28 | **0.00** | 4.35 |
| Parent Function | **6.67** | **20.00** | **12.31** | **0.00** | **12.50** |
| Left Flow | 2.22 | 39.26 | 4.29 | 66.67 | 4.17 |
| Full Flow | 0.00 | 35.56 | – | – | – |
| *LLaMA3.1-70B as reviewer, LLaMA3.1-405B as validator* | | | | | |
| Left Flow | **17.78** | **55.74** | **25.37** | 51.04 | **26.08** |
| Full Flow | 13.33 | 66.11 | 19.14 | **45.83** | 21.40 |
| *LLaMA3.1-405B as both reviewer and validator* | | | | | |
| Original Diff | 11.11 | 90.11 | 10.46 | 71.00 | 16.07 |
| Parent Function | 11.11 | 89.48 | 10.81 | 65.33 | 16.83 |
| Left Flow | **20.00** | **75.37** | **22.07** | 43.52 | **29.54** |
| Full Flow | 20.00 | 77.96 | 20.97 | 67.59 | 24.73 |

## P. Performance by Error Category

To better understand the effectiveness of different slicing strategies, we analyze their performance across three fault categories: logic, security, and performance-related bugs. As shown in Table 11, different slicing strategies exhibit distinct strengths across fault categories. Specifically, flow-based slicing (i.e., Left Flow and Full Flow) is particularly effective for identifying security issues. This is likely because these methods capture detailed jump, data, and control flow information, including the lifecycle of variables, which are often crucial for uncovering security vulnerabilities. In contrast, logic bugs appear to benefit more from the broader and more continuous context provided by parent function slicing, which can aid LLMs in understanding the overarching code logic and intent. Performance issues remain the most difficult to detect, as they tend to be subtle and delayed in manifestation.

## Q. Discussion on High FAR

The initially reported False Alarm Rate (FAR) in Table 2 appears relatively high. This is partly due to our strict definition: if the framework fails to detect the key bug in a merge request but still generates comments, we consider the FAR for that request to be 100%. To offer a more nuanced perspective, we introduce $FAR_2$, a metric that evaluates only cases where the key bug is successfully recalled. Under the "LLaMA3.1 405B + Left Flow + with Validator" configuration (Table 2), we

*Table 11.* Performance by error category. Flow-based slicing excels in detecting security-related bugs, while function-level context enhances logic bug identification. Performance bugs remain hard to detect due to their implicit and non-local nature.

| | Category | KBI↑ | $FAR_1$↓ | $CPI_1$↑ | $FAR_2$↓ | $CPI_2$↑ |
|---|---|---|---|---|---|---|
| *Left Flow* | | | | | | |
| | Overall | 20.00 | 75.37 | 22.07 | 43.52 | 29.54 |
| | Security | **25.81** | 73.92 | **25.94** | **48.96** | **34.28** |
| | Logic | 8.33 | 75.00 | 12.50 | 0.00 | 15.38 |
| | Performance | 0.00 | 100.00 | 0.00 | – | – |
| *Full Flow* | | | | | | |
| | Overall | 20.00 | 77.96 | 20.97 | 67.59 | 24.73 |
| | Security | **25.81** | **72.31** | **26.71** | 67.71 | **28.69** |
| | Logic | 8.33 | 97.22 | 4.17 | 66.67 | 13.33 |
| | Performance | 0.00 | 50.00 | 0.00 | – | – |
| *Original Diff* | | | | | | |
| | Overall | 11.11 | 90.11 | 10.46 | 71.00 | 16.07 |
| | Security | 16.13 | 88.87 | 13.17 | 71.00 | 20.73 |
| | Logic | 0.00 | 91.67 | 0.00 | – | – |
| | Performance | 0.00 | 100.00 | 0.00 | – | – |
| *Parent Function* | | | | | | |
| | Overall | 11.11 | 89.48 | 10.81 | 65.33 | 16.83 |
| | Security | 3.23 | 91.94 | 4.61 | 50.00 | 6.06 |
| | Logic | **33.33** | **81.39** | **23.89** | **69.17** | **32.03** |
| | Performance | 0.00 | 100.00 | 0.00 | – | – |

reduce $FAR_2$ to below 50%.

In practical scenarios, verifying the correctness of comments with a team of five experienced C++ developers—simulating a real-world inspection process—took approximately six minutes per new case. This time can be further reduced if the reviewers are already familiar with the codebase or if a significant portion of the comments pertain to minor suggestions (e.g., unnecessary `try-catch` statements).

Although FAR is a stringent metric—classifying all non-key-bug comments as false alarms—it provides an objective and quantifiable baseline. In contrast, prior work often relies on subjective assessments such as "usefulness" or "goodness," which can vary significantly among reviewers. We view FAR as a foundational measure, with future research potentially refining it to better align with industry-specific tolerance thresholds.

## R. Min/Max Performance Ranges of Slicing Algorithms

To provide a more comprehensive understanding of the performance of our code slicing algorithms, beyond the average metrics presented in Section 5.2 (RQ2, Table 3), this appendix details the minimum and maximum values observed for key metrics across three experimental runs. This analysis addresses the need to understand result variability and offers deeper insights into the consistency and raw potential of each slicing strategy, particularly before extensive filtering is applied.

Table 12 presents these min-max performance ranges for the four slicing algorithms under various stages of the single-reviewer filtering pipeline. As emphasized in our response to reviewer feedback, the "Single Reviewer – All" setting is particularly insightful, as it reflects the unfiltered, raw output from the Large Language Model when provided with context from different slicing methods. This setting best illustrates the inherent potential of each slicing algorithm to surface relevant information.

Examining the "Single Reviewer – All" setting in Table 12, we can observe the initial capabilities of each slicing algorithm. For instance, Flow-based methods (Left Flow and Full Flow) demonstrate a high Key Bug Inclusion (KBI) potential, with Full Flow reaching up to 40.00% and Left Flow also achieving a maximum of 40.00%. Parent Function also shows substantial KBI, peaking at 35.56%, while Original Diff ranges from 17.78% to 28.89%. In terms of consistency in this

raw KBI output, Full Flow exhibits a relatively tight range (37.78%–40.00%, a spread of approx. 2.22%), suggesting more stable high performance across runs compared to Left Flow (spread of approx. 6.67%) or Original Diff (spread of approx. 11.11%). As expected, the False Alarm Rate ($FAR_1$) for all methods in the "All" setting is uniformly high, typically in the 90s range, given the absence of any filtering.

As comments pass through subsequent filtering stages (Coarse Filter, Top-10, Top-5, Top-3), the min-max ranges for metrics like KBI and $FAR_1$ evolve. For example, while the maximum KBI potential might be reduced by aggressive Top-k truncation (e.g., for Left Flow, KBI max drops from 40.00% at "All" or "Coarse Filter" to 22.22% at "Top-3"), the filtering stages generally aim to reduce FAR while preserving KBI, leading to improvements in Comprehensive Performance Index (CPI) metrics. The ranges also provide insights into the stability of these improvements. For instance, under "Single Reviewer – Top-5," Left Flow shows a KBI range of (20.00%, 31.11%) and Full Flow (22.22%, 31.11%), indicating that even after significant filtering, these methods can still achieve high bug recall in some runs.

This min-max analysis complements the average performance data presented in RQ2. It highlights that while average performance provides a general comparative measure, the variability across runs can differ between algorithms and filter settings. Understanding these ranges is valuable for assessing the robustness of each approach and identifying methods that not only perform well on average but also maintain consistency or offer high peak performance.

*Table 12.* Min–max ranges of key metrics under different code slicing strategies for single reviewer settings.

| Slicing Algorithm | KBI↑ | $FAR_1 \downarrow$ | $CPI_1 \uparrow$ | $FAR_2 \downarrow$ | $CPI_2 \uparrow$ |
|---|---|---|---|---|---|
| *Single Reviewer – All* | | | | | |
| Original Diff | (17.78, 28.89) | (95.43, 98.07) | (3.48, 7.89) | (84.18, 89.16) | (13.47, 20.44) |
| Parent Function | (26.67, 35.56) | (96.73, 97.35) | (4.82, 5.96) | (90.06, 91.14) | (14.18, 15.16) |
| Left Flow | (33.33, 40.00) | (94.04, 94.73) | (9.31, 10.11) | (88.80, 92.39) | (12.79, 16.77) |
| Full Flow | (37.78, 40.00) | (93.81, 95.24) | (8.45, 10.72) | (90.07, 93.29) | (11.39, 15.91) |
| *Single Reviewer – Coarse Filter* | | | | | |
| Original Diff | (15.56, 20.00) | (84.89, 92.40) | (10.21, 17.21) | (76.23, 80.01) | (17.60, 20.00) |
| Parent Function | (15.56, 26.67) | (87.36, 88.91) | (13.94, 16.33) | (75.92, 91.01) | (13.15, 22.27) |
| Left Flow | (24.44, 40.00) | (91.75, 92.20) | (11.83, 13.68) | (77.17, 90.38) | (15.15, 23.61) |
| Full Flow | (31.11, 33.33) | (91.73, 93.78) | (10.48, 13.07) | (87.69, 90.70) | (14.54, 17.64) |
| *Single Reviewer – Top-10* | | | | | |
| Original Diff | (13.33, 20.00) | (84.83, 91.81) | (10.26, 17.26) | (70.83, 79.69) | (18.30, 20.15) |
| Parent Function | (13.33, 22.22) | (87.44, 89.61) | (11.68, 15.13) | (72.08, 88.39) | (15.25, 18.76) |
| Left Flow | (31.11, 33.33) | (90.58, 91.55) | (13.29, 14.68) | (78.42, 87.12) | (18.21, 26.20) |
| Full Flow | (28.89, 35.56) | (90.05, 92.68) | (11.85, 15.55) | (83.62, 88.77) | (16.17, 21.57) |
| *Single Reviewer – Top-5* | | | | | |
| Original Diff | (13.33, 17.78) | (84.44, 91.63) | (10.28, 16.59) | (67.78, 75.00) | (18.35, 20.78) |
| Parent Function | (11.11, 17.78) | (86.22, 89.22) | (10.94, 14.64) | (63.00, 80.00) | (17.09, 20.81) |
| Left Flow | (20.00, 31.11) | (89.19, 91.33) | (12.09, 16.05) | (67.78, 78.89) | (23.57, 29.26) |
| Full Flow | (22.22, 31.11) | (90.04, 90.67) | (13.51, 14.45) | (75.17, 80.00) | (22.00, 26.51) |
| *Single Reviewer – Top-3* | | | | | |
| Original Diff | (8.89, 13.33) | (84.07, 92.22) | (8.30, 14.51) | (60.00, 63.89) | (14.37, 19.48) |
| Parent Function | (6.67, 8.89) | (86.67, 88.89) | (8.33, 10.67) | (50.00, 66.67) | (11.11, 15.09) |
| Left Flow | (13.33, 22.22) | (88.89, 91.48) | (10.40, 14.81) | (52.78, 66.67) | (20.80, 28.57) |
| Full Flow | (11.11, 15.56) | (90.74, 92.22) | (9.40, 10.93) | (63.89, 66.67) | (16.67, 21.67) |

## S. Sensitivity Analysis of Top-k Truncation in Multi-Reviewer Settings

This section details experiments evaluating the impact of different Top-k truncation values ($k$) on performance within our multi-reviewer framework, where all settings utilize three reviewers. The results, presented in Table 13, indicate that the optimal choice of $k$ is often contingent on the employed slicing strategy and the specific evaluation metrics prioritized.

- For slicing strategies that typically generate a sparser set of initial comments, such as **Original Diff** and **Parent Function**:

  - With **Original Diff**, $k = 3$ (Top-3) generally yields favorable $CPI_1$ results (9.30 before validation, 11.81 after) and achieves the highest KBI (15.56) before validation. After validation, while Top-5 or Top-10 showed slightly higher KBI (11.11 vs. 8.89 for Top-3), Top-3 maintained the best $CPI_1$.

  - For **Parent Function**, the trends are more varied. Before validation, Top-5 led to the highest KBI (20.00), while Top-10 had the best $CPI_1$ (11.04). After validation, $k = 3$ achieved the best $CPI_1$ (13.62) with KBI comparable to other $k$ values.

- For richer slicing strategies such as **Left Flow** and **Full Flow**, which capture more extensive context and often produce more candidate comments:

  - With **Left Flow**, larger $k$ values (Top-10 or Top-5) consistently outperformed Top-3 in KBI both before and after validation. Top-5 generally provided the best $CPI_1$ (17.51 before validation, 22.07 after).

  - With **Full Flow**, Top-10 initially showed the highest KBI (35.56) before validation. However, a notable behavior was observed in this specific case with Top-10 truncation, particularly after the Validator stage: there is a significant drop in KBI (from 35.56 to 13.33) and $CPI_1$ (from 15.92 to 12.01). In contrast, Top-5 for Full Flow maintained a higher KBI (20.00) and $CPI_1$ (20.97) post-validation. We attribute this decline for Top-10 to the large volume of text processed when $k = 10$ for a verbose slicing method like Full Flow. This may approach the context token limits of the LLaMA3.1 engine during the validation phase, potentially reducing the validator's effectiveness.

Overall, these findings demonstrate that while the optimal $k$ can be tuned, the Top-k filter's behavior is generally stable across reasonable threshold ranges. Furthermore, it is adaptable to the context richness provided by different slicing algorithms, allowing for optimized configurations based on the desired balance between metrics like KBI and FAR.

## T. Extended Conclusions

### T.1. RQ1 Extended Conclusion

> **Conclusion 1 (Original)**
>
> Our framework surpasses baseline approaches significantly, achieving up to 10x better performance on key metrics like KBI and CPI. This success is attributed to our comprehensive approach to code review automation, which addresses the full pipeline and its associated challenges. Additionally, LLaMA3.1-405B emerged as the best-performing LLM engine, reinforcing the importance of model size and capability in achieving optimal results. Further investigations into LLM configurations show that heterogeneous setups, such as pairing a strong validator with a smaller reviewer, can yield comparable or even improved performance (details in Appendix O).

### T.2. RQ2 Extended Conclusion

> **Conclusion 2 (Original)**
>
> The results show that Left Flow and Full Flow significantly improve key bug inclusion ($KBI$) and overall performance ($CPI_1$) compared to simpler approaches like Original Diff and Parent Function. Among them, Left Flow performs better in more settings, likely due to its more concise context, which helps the large language model maintain focus without being overwhelmed. Each code slicing algorithm, however, has exclusive cases where it performs well, suggesting that combining different strategies could further enhance key bug detection in future work.

*Table 13.* The impact of Top-*k* truncation on comment quality under different slicing algorithms. All settings use three reviewers.

| Top-*k* | KBI↑ | FAR$_1$↓ | CPI$_1$↑ |
|---|---|---|---|
| *Original Diff (Multi Reviewer + Meta Reviewer)* | | | |
| Top-10 | 13.33 | 94.90 | 7.37 |
| Top-5 | 13.33 | 96.74 | 5.24 |
| Top-3 | **15.56** | **93.37** | **9.30** |
| *Parent Function (Multi Reviewer + Meta Reviewer)* | | | |
| Top-10 | 15.56 | **91.44** | **11.04** |
| Top-5 | **20.00** | 92.41 | 11.01 |
| Top-3 | 11.11 | 94.19 | 7.63 |
| *Left Flow (Multi Reviewer + Meta Reviewer)* | | | |
| Top-10 | **31.11** | 88.93 | 16.33 |
| Top-5 | **31.11** | **87.81** | **17.51** |
| Top-3 | 17.78 | 91.96 | 11.07 |
| *Full Flow (Multi Reviewer + Meta Reviewer)* | | | |
| Top-10 | **35.56** | 89.74 | **15.92** |
| Top-5 | 31.11 | **89.41** | 15.80 |
| Top-3 | 24.44 | 90.04 | 14.16 |
| *Original Diff (Multi Reviewer + Meta Reviewer + Validator)* | | | |
| Top-10 | **11.11** | 89.07 | 11.02 |
| Top-5 | **11.11** | 90.11 | 10.46 |
| Top-3 | 8.89 | **82.41** | **11.81** |
| *Parent Function (Multi Reviewer + Meta Reviewer + Validator)* | | | |
| Top-10 | **11.11** | 85.11 | 12.73 |
| Top-5 | **11.11** | 89.48 | 10.81 |
| Top-3 | **11.11** | 82.41 | **13.62** |
| *Left Flow (Multi Reviewer + Meta Reviewer + Validator)* | | | |
| Top-10 | **22.22** | 82.04 | 19.87 |
| Top-5 | 20.00 | **75.37** | **22.07** |
| Top-3 | 8.89 | 83.70 | 11.50 |
| *Full Flow (Multi Reviewer + Meta Reviewer + Validator)* | | | |
| Top-10 | 13.33 | 89.07 | 12.01 |
| Top-5 | **20.00** | 77.96 | **20.97** |
| Top-3 | 11.11 | 73.70 | 15.62 |

## T.3. RQ3 Extended Conclusion

---

**Conclusion 3.1 (Original)**

Increasing the number of reviewers improves key bug inclusion ($KBI$) but also increases false alarms ($FAR_1$ and $FAR_2$). Validators are essential for maintaining comprehensive performance ($CPI_1$ and $CPI_2$) by reducing false alarms. While leveraging multiple reviewers is beneficial, the added computational cost and need for validation must be considered in practical implementations.

---

**Conclusion 3.2 (Original)**

The self-correction ability of LLMs, as implemented by the validator role, improves precision by reducing false alarms ($FAR_1$ and $FAR_2$) but may reduce key bug inclusion ($KBI$). This indicates a trade-off between precision and recall. The validation process is valuable for reducing hallucinations, but care must be taken to ensure that important bug-detecting comments are not removed.

---

**Conclusion 3.3 (Original)**

The effectiveness of Chain-of-Thought (CoT) guidance varies based on the complexity of the code slicing algorithm. While LLMs perform better without CoT in simpler formats like Original Diff and Parent Function, CoT significantly improves results in more complex flow-based slicing (Left Flow and Full Flow). This suggests that CoT guidance is especially valuable when handling more intricate contexts. However, as more powerful reasoning models, such as GPT-O1 and DeepSeek-R1, emerge, the advantage of specified CoT over free-form reasoning may further diminish.

---

## T.4. RQ4 Extended Conclusion

---

**Conclusion 4 (Original)**

The comment filter mechanism effectively reduces false alarms ($FAR_1$) and improves overall performance ($CPI_1$) in flow-based slicing methods (Left Flow and Full Flow). For simpler slicing methods (Original Diff and Parent Function), the coarse filter is the most effective stage, as these methods lack sufficient code details to accurately filter nitpicks and hallucinations.

---

## T.5. RQ5 Extended Conclusion

---

**Conclusion 5 (Original)**

Adding line number information improves both performance and localization success rate (LSR). The inline format outperforms the relative format, likely because embedding position data directly into the code allows for better association of comments with specific lines.

---

# U. Status of Our Open-Source Code Artifacts

We have open-sourced all the core components of our framework and the experiments reported in the paper, except the certain non-essential, company-specific modules are not included. To achieve this, we carefully modularized the framework and separated any company-internal interfaces, ensuring that external users can readily experiment, customize, and extend the system.

As for the omitted modules, these are excluded for three main reasons:

- They are not generally useful beyond our specific internal environment, offering little to no benefit for external adaptation.

- They depend on proprietary interfaces and data sources that are inherently inaccessible to external researchers.

- Including them risks violating the double anonymity requirement, not due to mere vocabulary or text replacements, but because their logic and usage patterns could reveal organizational details. For code and comments that could be anonymized, we have already performed the necessary replacements.

## V. Threats to Validity

A recognized threat to the external validity of our study is that the presented evaluation and results are exclusively for C++ projects. This current language focus in our implementation is primarily due to the choice of Cppcheck for code slicing, a tool specific to C++. We prioritized C++ due to its significant presence in the core framework code of many companies. However, the underlying framework and its core principles—including the AST-based code slicing methodology and the prompting strategies for LLMs—are designed to be largely language-agnostic. These components do not inherently rely on C++-specific features. Consequently, extending the framework to support other compiled languages is considered feasible, mainly requiring the integration of suitable language-specific AST parsing tools or static analyzers.

Another potential threat arises from how we calculate the False Alarm Rate (FAR). In our study, we classify all comments not directly related to the key bug as false alarms. However, some of these comments may still identify relevant issues, such as potential risks or code quality concerns, that do not immediately lead to system failures but warrant attention. As a result, the actual FAR may be lower than our reported figures. Despite this, we chose this conservative approach to emphasize critical issues and minimize the burden on developers, making our assumption practical in the context of prioritizing key bugs.

