# OpenReview forum: "Towards Practical Defect-Focused Automated Code Review"
_ICML.cc/2025/Conference — ICML 2025 spotlightposter_

### Official Review · Reviewer_ngPf · 2025-02-14

**Overall Recommendation:** 4

**Summary:**

The paper introduces an end‐to‐end automated code review system designed specifically for defect detection in large-scale, industrial codebases. The authors identify four key challenges in automating code review: capturing the full, relevant code context; improving key bug inclusion to ensure that critical defects are detected; reducing the false alarm rate by filtering out redundant or irrelevant comments; and integrating the system into human workflows.

To overcome these challenges, the paper proposes an approach that includes:
• A static analysis system using several code slicing algorithms
• A multi-agent framework where different LLM roles (Reviewer, Meta-Reviewer, Validator, and Translator) collaborate
• A robust filtering mechanism that systematically reduces false positives by eliminating nitpicks and hallucinations
• A line-aware prompt design that ensures review comments are accurately attached to the relevant lines of code

Empirical evaluations demonstrate that this integrated approach can achieve good performance.

**Claims And Evidence:**

Most of the paper's claims are supported by empirical evaluations and ablation studies.
The evidence might be less convincing on the generalizability of the approach to other programming languages.

**Essential References Not Discussed:**

N.A.

**Experimental Designs Or Analyses:**

One potential issue is that while these metrics and the accompanying analyses are well-structured, some evaluation criteria (like KBI and CPI) are specific to this work and might not fully capture all dimensions of code review quality as experienced by developers and can not be well compared with the literatures..

**Methods And Evaluation Criteria:**

The methods and evaluation criteria proposed in the paper are well-aligned.
On the evaluation side, the authors introduce metrics such as Key Bug Inclusion (KBI), False Alarm Rate (FAR), Comprehensive Performance Index (CPI), and Line Localization Success Rate (LSR).

**Other Comments Or Suggestions:**

N.A.

**Other Strengths And Weaknesses:**

Strengths:

The paper presents an innovative integration of multiple techniques—including diverse code slicing algorithms, a multi-agent framework with chain-of-thought reasoning, and a robust filtering mechanism—to tackle real-world challenges in automated code review.
The experimental evaluation is comprehensive and grounded in industrial-scale data, using metrics (KBI, FAR, CPI, LSR) that directly address practical defect detection concerns.

Weaknesses:

Some design choices, such as the thresholds in the filtering mechanism, appear heuristic, and it remains unclear how sensitive the results are to these parameters.
The evaluation is primarily confined to a specific industrial C++ codebase, leaving questions about the generalizability of the approach to other programming languages or domains.

**Questions For Authors:**

1. Could you elaborate on the trade-offs observed with the validator role?

2. Could you discuss how your approach might generalize to other programming languages or domains?

3. How sensitive is the system’s performance to the thresholds, and have you considered adaptive or data-driven methods for threshold selection?

4. Have you conducted any user studies or qualitative evaluations with developers to validate that these metrics align well with real-world expectations of code review quality?

**Relation To Broader Scientific Literature:**

The key contributions of the paper extend prior findings to an integrated, context-rich, and workflow-aware system that is validated on industrial-scale data.

**Theoretical Claims:**

The paper does not include formal proofs or theoretical claims.

---

> ### Author Rebuttal · Authors · 2025-04-01
>
> **We sincerely thank the reviewer for the thoughtful and encouraging feedback. We greatly appreciate your recognition of the system design, the practical orientation of our evaluation, and the paper’s contributions toward large-scale industrial defect detection. Below, we address your questions in turn.**
>
> ---
>
> ### **1. Trade-offs with the Validator Role**
>
> **(Response to “Question 1”)**
>
> The Validator role involves important trade-offs:
>
> - It is highly effective at removing hallucinated comments, which was a core motivation for introducing it.
> - However, we observed some negatives cases, where valid comments were mistakenly rejected. These cases typically stem from:
>   - **Context loss** during the Meta-Reviewer phase (e.g., mismatched file names or snippet IDs), which prevented the Validator from tracing back to the correct source code.
>   - **Position mismatches**, where a comment was attached to a nearby but unrelated line.
>   - **Token limit issues**, where long inputs (context + chain-of-thought + comment pairs) exceeded the model’s context window.
>   - **Score variance**, where randomly low Q1–Q3 scores led to inadvertent filtering.
>
> We believe that introducing more robust communication between roles and improving traceability can reduce such errors. This is a promising direction for future work.
>
> ---
>
> ### **2. Generalization to Other Languages or Domains**
>
> **(Response to “Question 2”)**
>
> Our system is **inherently language-agnostic**. It relies on structural elements like **function boundaries, control/data flow, and AST-derived slicing**, which are common across many programming languages.
>
> We chose C++ due to its industrial relevance and high complexity. However, both our slicing algorithms and prompting strategies are adaptable to other languages. Extending the framework to support multiple languages is part of our planned roadmap.
>
> ---
>
> ### **3. Sensitivity to Filtering Thresholds**
>
> **(Response to “Question 3”)**
>
> The thresholds for Q1 (nitpicks) and Q2 (hallucinations) were set heuristically for interpretability and practical deployment. As described in Section 3.4, we adopted a 1–7 scoring scale (inspired by McAleese et al., 2024), with a threshold of 4+ to denote serious, actionable issues.
>
> We evaluated Q3 (redundancy) sensitivity via a top-*k* truncation strategy in **Section 5.4**, and extended it further:
>
> 🔗 https://anonymous.4open.science/r/16368/The%20impact%20of%20Top-k%20truncation.pdf
>
> Key observations:
>
> - **Top-3** truncation works better for **shorter context slices** (e.g., Original Diff, Parent Function), which produce fewer relevant comments.
> - **Top-5 to Top-10** perform better with **richer slicing methods** (e.g., Left Flow, Full Flow), which generate more detailed reviews.
> - **Full Flow + Top-10** occasionally shows performance drop after validation, likely due to hitting the token limit of LLaMA3.1, reducing effectiveness.
>
> We agree that adaptive or learned thresholds would be valuable. However, to avoid overfitting to a specific codebase, we chose to retain a generalizable, interpretable heuristic strategy in this version. We will reflect this design choice and its limitations more clearly.
>
> ---
>
> ### **4. Alignment of Metrics with Real-World Review Quality**
>
> **(Response to “Question 4”)**
>
> While we have not yet conducted formal user studies, our metrics (KBI, FAR, LSR) are grounded in interviews with professional developers. Developers repeatedly emphasized two points:
>
> 1. **Catching critical bugs** is a top priority.
> 2. **Reducing irrelevant comments** is crucial for adoption—"even one false positive can erode trust."
>
> This informed our focus on *key defect inclusion* and *false alarm suppression*, which are not well captured by traditional text similarity metrics.
>
> The system is currently deployed in an internal development team, though we do not require developers to reply to each comment. Based on initial feedback, FAR has emerged as a particularly sensitive metric, directly affecting developer perception. We agree that systematic user studies would further strengthen the validation and plan to incorporate them in future work.
>
> ---
>
> **We thank the reviewer again for the constructive questions and positive evaluation. We hope these clarifications further highlight the robustness and extensibility of our work.**

---

> > ### Comment · Reviewer_ngPf · 2025-04-02
> >
> > I am fine with the answers.

---

### Official Review · Reviewer_ysLJ · 2025-02-18

**Overall Recommendation:** 4

**Summary:**

The paper presents a language-model based system for automated code review. The methodology boils down to using static analysis tools to identify the most relevant parts of the code base, and then passing this through several LLM calls to generate the review, identify the key components, and filter out noise. The authors state that this approach is inspired by interviews with real-world code reviewers. Empirically, the authors find that on a dataset consisting of merge requests from an internal base, their method outperforms prior work. In particular, they evaluate their method in terms of whether the review correctly identifies the key bug introduced in the merge request, as well as how many "false alarms" the review raises and whether the faulty line(s) are correctly identified.

## update after rebuttal
Following the end of the rebuttal period I have decided to further raise my score to Accept (4). I like the motivations of the work and think it could have real-world impact, especially as C++ is at this point criminally understudied by academia in comparison to its popularity in the industry. The uncertainty quantification included by the authors in their final reply has convinced me that the findings are meaningful, and I appreciate that they have been responsive to changes requested in both the scope of the work as well as its exposition.

I will note that I am not sure if ICML is the right venue for this work, as opposed to conferences such as ICSE or FSE, but given that none of the other reviewers seemed bothered by this I will chalk it up to being a symptom of AI/ML becoming increasingly relevant to many other areas of computer science and will not hold it against this paper in particular.

**Claims And Evidence:**

Overall, the claims made appear to be well supported by the experiments. Of course it is difficult to judge the validity of evaluating on an internal dataset, but I do not see any immediate causes for concern. The one thing that stands out to me as perhaps not being completely justified by the experiments is whether the method reduces the False Alarm Rate (FAR); the definition of this metric appears to be quite strict, as acknowledged by the authors in Appendix N and Q. Furthermore, it seems to go against the training objectives of prior work like CodeReviewer, so the fact that this new method outperforms it on this metric seems like an obvious result. However, this limitation is acknowledged by the authors (albeit hidden in the appendix), and if prior work failed to take real-world concerns such as limiting the cognitive burden imposed on the user then that is a shortcoming of their approach, not of this paper.

**Essential References Not Discussed:**

None that I am aware of; this problem is well-studied in the software engineering literature, but the authors appear to have cited the most relevant works already (in particular those by Tufano et al.). I was surprised by the authors' claim in Appendix D that slicing (which is a very well established technique across software engineering) had not yet been applied to automated code review, but could not find any references myself for this, so it may be true.

**Experimental Designs Or Analyses:**

I have checked the soundness of the experiments presented in the main text, and they appear valid and sound. The code slicing experiment in 5.2 seems a bit weak, however, since the absolute numbers of key bugs is relatively small, so it is hard to tell whether the different spreads in the Venn diagram are significant or just due to chance. Another, more important issue is that the text mentions that each experiment is repeated 3 times to account for the stochasticity of the system, but as far as I can see the results of each experiment is just reported as the mean of these 3 runs, rather than their complete spread. Just reporting the min and the max of each number would already be a significant improvement in terms of making it easier to tell if the results are significant or not.

**Methods And Evaluation Criteria:**

Yes, the internal dataset used for the evaluation appears to be well suited for the study.  As mentioned in the preceding section, the FAR metric appears to be somewhat arbitrary. In particular, having a code review system point out minor issues and not just major faults seems like a feature, not a failure, to me. However, I appreciate that the authors are attempting to limit the amount of information that is presented to the end-user, since this is likely a make-or-break factor when applying the system in the real world.

**Other Comments Or Suggestions:**

Section 6 uses a very strange reference format, which should be simplified. For example, "Gupta et al. (Gupta & Sundaresan, 2018)" should just be "Gupta & Sundaresan (2018)".

I really disagree with framing your system as being "multi-agent". There is no interaction with an environment, so there is in fact no agent involved at all. This is an abuse of the nomenclature.

**Other Strengths And Weaknesses:**

Strengths:
- The paper is well organized, with clear hypotheses and thorough discussion of the experiments
- The authors release the source code of their system, which will help others seeking to replicate or build upon their work

Weaknesses:
- The figures and tables are, on the whole, completely illegible (at least when printed). The text is much too small (in particular in Figures 1,2, 3, 4) and the black-on-dark-green formatting of the tables is difficult to read.
- While very much understandable, it is unfortunate that the dataset could not be released, as this would certainly have been a very useful resource for others working in this area.

**Questions For Authors:**

1. What was your motivation for submitting this work to ICML instead of ICSE, FSE or ASE, where the majority of prior work in this area was published?
2. How certain are you that there is no prior work applying slicing in the context of automated code review?

**Relation To Broader Scientific Literature:**

Automated code review is a problem with significant real-world impact, and has been studied in the software engineering community for many years. This paper appears to be a significant step forward in this domain, since it evaluates the performance of their system on real-world fault reports and merge requests, using metrics that (are intended to) mimic the desiderata of real software developers. The methodology is not itself particularly novel or interesting, but the experiments are extensive and may inspire future work in this direction. I am not sure how interesting this paper would be to an ICML audience, rather than a software engineering audience, though.

**Theoretical Claims:**

N/A; no theoretical claims.

---

> ### Author Rebuttal · Authors · 2025-04-01
>
> **We sincerely thank the reviewer for the constructive and thoughtful feedback. We appreciate your recognition of the clarity of the paper, the strength of the experimental design, and the potential real-world impact of our work. Below, we respond to your questions and key suggestions.**
>
> ---
>
> ### **1. Motivation for Submitting to ICML**
>
> **(Response to “Questions for Authors”)**
>
> We submitted this work to ICML because we believe that integrating large language models (LLMs) into software engineering workflows represents a growing and impactful frontier at the intersection of machine learning and software development.
>
> With the rapid advancement of LLMs, core tasks in software engineering—such as code review automation—are approaching an inflection point. Engaging the ML community is essential to foster deeper cross-domain innovation, especially as challenges like contextual reasoning, modular prompting, and learning from structured data remain open research problems.
>
> Moreover, our defect-focused formulation reflects a broader ML interest in real-world, system-level tasks beyond conventional generation. Practically, we also note the limited availability of top-tier SE venues with winter deadlines.
>
> ---
>
> ### **2. Novelty of Code Slicing in Code Review Context**
>
> **(Response to “Questions for Authors” & “Essential References”)**
>
> To the best of our knowledge, our work is the first to explicitly incorporate **static code slicing** into an **LLM-based automated code review pipeline** to guide review comment generation and filtering.
>
> While slicing is a well-established technique in software engineering, prior works on code review automation have largely focused on snippet-level generation or comment naturalness, rather than **context-aware defect localization**. Under such task formulations, slicing has typically been overlooked, as the repository-level context is not utilized.
>
> We acknowledge this is a strong claim and will revise the relevant statement in **Appendix D** to present it more cautiously.
>
> ---
>
> ### **3. Use of the Term “Multi-Agent”**
>
> **(Response to “Other Comments”)**
>
> We appreciate this observation. Our usage of “multi-agent” was intended to convey modular collaboration among LLM roles (e.g., Reviewer, Validator, Aggregator), not reinforcement learning-style agents interacting with an environment.
>
> To avoid confusion, we will revise the terminology in the revised version to use terms such as **“multi-role framework”** or **“role-based architecture.”**
>
> ---
>
> ### **4. Additional Suggestions and Minor Corrections**
>
> - **Figures and tables** will be updated with larger fonts, improved contrast, and clearer layout to ensure readability in both digital and printed formats.
> - **Citation formatting** in Section 6 will be corrected as suggested (e.g., “Gupta & Sundaresan (2018)”).
>
> ---
>
> **We thank the reviewer again for the generous feedback and helpful suggestions. We hope these clarifications address your concerns and reinforce the value of our contribution.**

---

> > ### Comment · Reviewer_ysLJ · 2025-04-02
> >
> > Thank you, authors, for your clear and concise response. I am happy with the changes you have outlined and will consider updating my score.
> >
> > One final question I have is whether your updated tables will include (min, max) ranges (over the 3 runs), rather than just the mean, as requested in my review? I did not see this mentioned in your response. I think quantifying the uncertainty involved in your experiments is essential so that the readers of the paper can gain some confidence about whether your results are statistically significant. A 95% CI on the means or something like that would have been ideal but such an interval would be difficult to construct with only 3 samples, so reporting (min, max)-tuples will be sufficient in this case. If you believe this would clutter the tables then at least include it in the appendix.
> >
> > In your reply, if possible, it would be good to include such a table, since if the ranges overlap significantly between your method and the baselines then I would have to reconsider my score on the basis of there being insufficient evidence for your claims.

---

> > > ### Author Response · Authors · 2025-04-04
> > >
> > > **We sincerely thank the reviewer for the thoughtful feedback and for acknowledging our changes. We are glad that the updates align with your expectations. Regarding the additional question, we have provided the following clarifications.**
> > >
> > > ---
> > >
> > > We fully agree with the reviewer that quantifying **uncertainty** in experimental results is essential for building confidence in the findings. We appreciate your input on this matter.
> > >
> > > To address this, we include **min/max ranges** comparing the performance of our system with the baselines, allowing readers to see the variability across different runs. We provide this data in the updated tables:
> > >
> > > 🔗 [Min/Max ranges for comparing baselines](https://anonymous.4open.science/r/16368/Comparing%20baselines.pdf)
> > >
> > > These results show that our workflow significantly outperforms baseline approaches, even considering experimental uncertainty. This success highlights the effectiveness of our approach to end-to-end automation, due to its pipeline design and alignment with developers' expectations.
> > >
> > > Furthermore, as the reviewer emphasized the importance of the slicing algorithms, we have also reported the **min/max ranges** of the comparison for our **slicing algorithms**:
> > >
> > > 🔗 [Min/Max ranges for slicing algorithm comparison](https://anonymous.4open.science/r/16368/Max%20min%20slicing.pdf)
> > >
> > > We recommend that the reviewer pay special attention to **single reviewer settings**, as these are closest to the raw output and highlight the potential of the slicing algorithms before applying our filtering mechanisms. For example, the "Single Reviewer – All" setting presents the raw output before filtering is applied.
> > >
> > > Additionally, to address concerns about the Venn diagram analysis and the possibility of chance-based spread, we have included **results across different fault categories**:
> > >
> > > 🔗 [Fault category-based analysis](https://anonymous.4open.science/r/16368/Performance%20by%20error%20category.pdf)
> > >
> > > Our findings indicate that **flow-based slicing** particularly benefits the identification of **security errors**, as it captures **jump, data, and control flow**, including the lifecycle of variables. In contrast, **parent function slicing**, which provides broader and more continuous context, helps the LLMs understand **code logic**, leading to better performance on **logic errors**.
> > >
> > > ---
> > >
> > > **We thank you again for your detailed feedback. We believe these updates will significantly improve the presentation of our results and provide the necessary confidence in our findings.**

---

### Official Review · Reviewer_6FuX · 2025-03-14

**Overall Recommendation:** 3

**Summary:**

This paper proposes an advanced method for automating code reviews, focused on defect detection and improving real-world code review workflows. To address the challenges, the authors introduce a multi-agent LLM framework that utilizes code slicing algorithms, a filtering mechanism to remove irrelevant comments, and a new prompt design for better integration into human workflows. Their system was validated using real-world industry data, achieving significant improvements over previous methods in detecting key bugs, reducing false alarms, and improving code review performance.

**Claims And Evidence:**

1. The evaluation primarily focuses on performance metrics rather than directly on qualitative assessments and user studies that demonstrate improved usability or reduced developer burden. Thus, the claim "Real-world Workflow Integration" is limitedly supported.
2. The title didn't specifically mention C++ focus, but the "10× improvement" is primarily based on C++ code. This raises the concern about the generality of the framework, and the scope of the paper

**Essential References Not Discussed:**

No

**Experimental Designs Or Analyses:**

The paper does not provide an in-depth justification for the chosen thresholds of the redundancy comment filter. A sensitivity analysis regarding these thresholds could strengthen confidence in the filtering mechanism's robustness.

**Methods And Evaluation Criteria:**

Assuming the paper focuses on C++ related code review, the methods and evaluation make sense.

**Other Comments Or Suggestions:**

N/A

**Other Strengths And Weaknesses:**

### Strengths

1. The use of multiple agents adds flexibility and scalability, ensuring that the review process is thorough and precise.

2. The method achieves up to 10x improvement over previous baselines in detecting critical bugs and reducing irrelevant comments.


### Weaknesses

1. The framework is mainly tailored for C++, and its adaptability to other languages may need further research and development.

2. The effectiveness of the framework strongly depends on the underlying LLM engine, with larger models like LLaMA3.1 performing better, which might be a limitation for certain applications where computational resources are constrained.

3. The more detailed slicing methods (like Left Flow and Full Flow) showed better performance but could potentially be computationally expensive or difficult to implement in all environments.

**Questions For Authors:**

Please refer to weaknesses.

**Relation To Broader Scientific Literature:**

The paper extended existing automated code review methods from isolated snippet-level generation to holistic, context-rich analysis. And by proposing a multi-agent, collaborative framework with evaluation metrics that better capture the practical realities of defect detection. These innovations build directly on and address limitations identified in prior work.

**Theoretical Claims:**

The paper primarily focuses on designing and empirically evaluating its defect-focused automated code review framework.

---

> ### Author Rebuttal · Authors · 2025-04-01
>
> **We thank the reviewer for the constructive and insightful feedback. We appreciate your recognition of our system’s architecture and the observed performance improvements on real-world data. Below, we address your concerns regarding generality, LLM dependency, filter sensitivity, and workflow integration.**
>
> ---
>
> ### **1. Generality Beyond C++**
>
> **(Response to “Weakness 1”)**
>
> > The framework is mainly tailored for C++, and its adaptability to other languages may need further research and development.
>
> We agree clarification is needed. The current results are based on C++ and the 10× improvement reflects this. We will state this explicitly in the revised abstract. As noted in **Appendix Q**, we chose C++ because:
>
> - It is one of the most widely used industrial languages;
> - It is among the most complex in mainstream use.
>
> Nonetheless, our framework is **inherently language-agnostic**, relying on universal structures like ASTs to extract elements such as control/data flow and function scopes. These structures exist across most compiled languages, and the framework’s design—including slicing and prompting—does not rely on C++-specific features. Extension to other languages is thus feasible and ongoing work.
>
> ---
>
> ### **2. LLM Dependency and Computational Cost**
>
> **(Response to “Weakness 2, 3”)**
>
> > The framework depends on large LLMs... Detailed slicing may be computationally expensive...
>
> We acknowledge that performance depends on model capability. While we use large models (e.g., LLaMA 3.1) in our main experiments, we also evaluated smaller models in **Section 5.1 (RQ1)**. These results show that compact models with strong reasoning abilities can still perform competitively, especially there is a trend towards increasing capacity density.
>
> For slicing cost: while fine-grained slicing (e.g., Full Flow) is more expensive, it remains within practical limits. We report detailed runtimes via violin plots:
>
> 🔗 https://anonymous.4open.science/r/16368/Runtime%20violin%20plot.pdf
>
> - **Median runtime per MR** is **6.2 minutes**.
> - This fits comfortably within typical CI/CD pipelines (15–30 minutes), which include compilation, linting, static analysis, testing, and deployment checks.
>
> Our system runs **in parallel** with these processes and does not introduce blocking delays. We believe this makes the cost acceptable for real-world use. Further analysis is included in **Appendix M**.
>
> We also emphasize that our focus is on solving core challenges like *defect inclusion* and *false alarm suppression*. Cost-efficiency optimization (e.g., model scaling, caching) is important but future work.
>
> ---
>
> ### **3. Sensitivity Analysis of Filter Thresholds**
>
> **(Response to “Experimental Designs”)**
>
> > The paper does not provide an in-depth justification for the chosen thresholds of the redundancy comment filter...
>
> Thank you for this suggestion. Our current thresholds are based on heuristic rules aimed at **interpretability**. For Q1/Q2, a score above 4 on a 1–7 scale (inspired by prior work) indicates actionable, serious issues. This setting aligns with developer feedback during internal piloting.
>
> For Q3, we conduct a sensitivity study using top-*k* truncation in **Section 5.4 (RQ4)**. We now extend this to multi-reviewer settings:
>
> 🔗 https://anonymous.4open.science/r/16368/The%20impact%20of%20Top-k%20truncation.pdf
>
> Key observations:
>
> - **Smaller top-*k*** (e.g., 3) performs better for shorter slicing strategies (Original Diff, Parent Function), where fewer high-quality comments exist.
> - **Larger top-*k*** (5–10) works better for richer slicing (Left Flow, Full Flow), which yields more relevant outputs.
> - For Full Flow with Top-10, performance declines after validation—likely due to hitting token limits, reducing model focus.
>
> These findings demonstrate that the filter behavior is stable across reasonable thresholds, and adaptable to context richness.
>
> ---
>
> ### **4. Real-World Workflow Integration**
>
> **(Response to “Claims And Evidence”)**
>
> > The claim “Real-world Workflow Integration” is limitedly supported.
>
> Thank you for pointing this out. The workflow is illustrated in **Figure 1**, but we agree it merits a more detailed explanation. We will provide a full description in an additional appendix.
>
> Briefly, when a merge request is submitted:
>
> 1. The system verifies user and file access;
> 2. Slicing and multi-agent review are launched;
> 3. Filtered comments are injected into the internal DevOps system;
> 4. Comments are positioned at exact line numbers and pushed to developers via messaging.
>
> This enables seamless integration into daily development workflows. In particular, **line-aware comment injection**, evaluated in **Section 5.5 (RQ5)**, was critical for developer adoption and feedback.
>
> ---
>
> **We will incorporate these clarifications and additions in the revised version. We hope this addresses your concerns and supports a more favorable assessment. Your feedback has helped improve both clarity and rigor.**

---

### Official Review · Reviewer_jwAu · 2025-03-16

**Overall Recommendation:** 2

**Summary:**

In this paper, the authors proposed a framework for automated code review. More specifically, the authors first used code slicing to enable the Multi-Agent Code Review System to obtain sufficient context fragments of the code. Then, the Multi-Agent Code Review System conducted code reviews, filters, aggregates, and ranks the reviews. Finally, the proposed approach localized the issues mentioned in the reviews. The authors created a dataset at the merge request level using data from four repositories. The authors experimentally validated the performance  using Key Bug Inclusion, False Alarm Rate, Comprehensive Performance Index, and Line Localization Success Rate. The proposed approach performed better on both the Left Flow and Full Flow slicing algorithms, showing effective localization capability in the Inline representation.

**Claims And Evidence:**

The authors pointed out that current evaluations rely excessively on textual similarity metrics (e.g., BLEU, ROUGE), which fails to measure real-world effectiveness. Is there a connection between metrics like BLEU and the evaluation metrics used by the authors? The authors could compare the proposed approach and baselines with BLEU and ROUGE.

**Essential References Not Discussed:**

There are some more recent related work. For example:

[1] Wang L, Zhou Y, Zhuang H, et al. Unity Is Strength: Collaborative LLM-Based Agents for Code Reviewer Recommendation[C]. IEEE/ACM International Conference on Automated Software Engineering. 2024: 2235-2239.

[2] Wei Tao, Yucheng Zhou, et al., 2024. KADEL: Knowledge-Aware Denoising Learning for Commit Message Generation. ACM Trans. Softw. Eng. Methodol. 33, 5, Article 133 (June 2024), 32 pages.

[3] Yu Y, Rong G, Shen H, et al. Fine-tuning large language models to improve accuracy and comprehensibility of automated code review[J]. ACM transactions on software engineering and methodology, 2024, 34(1): 1-26.

**Experimental Designs Or Analyses:**

The authors evaluated the performance of different LLMs; however, further analysis was needed to examine the performance differences resulting from various combinations of LLMs. Additionally, did different tasks yield performance variations when different LLMs were employed? For instance, using LLaMA 3.1 (405B) as the Reviewer and LLaMA 3.1 (70B) as the Validator. For Time Cost in Appendix M, Experimental Setups in Detail, the authors mentioned that one round of automated code review took approximately 9 hours. In the context of code review, were such time and resource overheads acceptable?

**Methods And Evaluation Criteria:**

The authors have established criteria based on key issues; however, it is unclear how the key issues are defined. Is there a specific set of evaluation criteria, and are these standards reasonable? Regarding the Dataset, a C++ dataset was conducted by using requests. The authors categorized errors into logic errors, code security errors, and performance-related errors. Does the dataset comprehensively cover the majority of error types encountered in C++? Is it feasible to report the performance of the proposed method across these different error categories?

**Other Comments Or Suggestions:**

Section 2.1 is mentioned in Appendix A, but where is it in the paper?

**Other Strengths And Weaknesses:**

Strengths:
1. The proposed merge-request-based code review could be useful for real-world applications.
2. The authors introduce code localization based on the review, which could help quickly determine whether the issues are genuine.

Weakness:
1. The proposed approach does not validate the combination of different LLMs.
2. The authors do not discuss whether the time and resource costs of the multi-agent code review system are acceptable.
3. In the LLM prompts, the authors can discuss the effectiveness of Retrieval-Augmented Generation in LLM-generated outputs.
4. The authors evaluate the comments based on certain issues. However, the evaluation should be improved. For instance, the authors should assess whether the answers are vague.

**Questions For Authors:**

Can the authors provide more information about the dataset? Does the dataset comprehensively cover the majority of error types? How were the 45 fault reports selected? Are they representative of common code review scenarios (e.g., edge cases vs. typical defects)?

**Relation To Broader Scientific Literature:**

N/A

**Theoretical Claims:**

The paper does not contain proofs for theoretical claims.

---

> ### Author Rebuttal · Authors · 2025-04-01
>
> **We thank the reviewer for the constructive comments and the recognition of our framework’s practical impact. We have addressed the concerns by enhancing dataset transparency, adding new evaluations (including error category analysis and heterogeneous LLM roles), clarifying metric rationale, and updating implementation files.**
>
> ---
>
> ### **1. Dataset Scope and Fault Selection**
>
> **(Response to “Questions for Authors” and “Methods and Evaluation Criteria”)**
>
> > Can the authors provide more information about the dataset?...
> > It is unclear how the key issues are defined... Is it feasible to report the performance across different error categories?
>
> We appreciate your interest in the dataset. To improve transparency, we added a desensitized JSON folder of fault descriptions to our updated Zenodo repository and will revise **Appendix J** accordingly.
>
> **Fault selection** follows a practical criterion: all faults caused user-visible issues and were formally logged in the company’s internal defect tracking system (Section 3.6). This **result-oriented** strategy emphasizes **real impact**, even if it doesn’t fully cover all C++ error types.
>
> The dataset includes both edge and typical cases, e.g.:
>
> - **Case 4694_23117**: array out-of-bounds and null-pointer dereference.
> - **Case 16231_13308**: misuse of `boost::random::beta_distribution`.
>
> To analyze per-category performance, we added a breakdown across **logic**, **security**, and **performance-related** bugs:
>
> 🔗 https://anonymous.4open.science/r/16368/Performance%20by%20error%20category.pdf
>
> Findings show that flow-based slicing benefits security bugs, while broader context helps with logic bugs.
>
> ---
>
> ### **2. Evaluation Metrics and Comment Quality**
>
> **(Response to “Claims and Evidence” and “Weakness 4”)**
>
> > The authors pointed out that current evaluations rely excessively on BLEU, ROUGE...
> > The authors evaluate the comments based on certain issues... assess whether the answers are vague.
>
> We intentionally did **not** use BLEU or ROUGE due to several limitations:
>
> 1. Our task involves **many-to-many** mappings between code and reviews, violating BLEU’s single-reference assumption.
> 2. Code review requires reasoning and domain expertise, and recent studies from earlier this year show that BLEU and ROUGE fail to capture quality effectively in such tasks.
> 3. Real fault reports and LLM outputs differ significantly in style and expression, making textual similarity unreliable.
>
> On vagueness: rather than evaluating writing style, we focus on key outcome metrics like *key bug inclusion* (KBI) and *false alarm rate* (FAR), which directly reflect review effectiveness. These metrics are more objective and interpretable and will be further discussed in **Appendix N**.
>
> ---
>
> ### **3. LLM Combinations and Time Cost**
>
> **(Response to “Experimental Designs or Analyses” and “Weakness 1, 2”)**
>
> > The proposed approach does not validate different LLM combinations...
> > The authors do not discuss whether the time and resource costs are acceptable...
>
> Our main experiments use the same LLM across all agents to isolate whether a **strong model** alone can resolve key challenges in code review. However, we agree that heterogeneous combinations are worth exploring.
>
> We added experiments varying reviewer/validator assignments, showing that a strong validator paired with a smaller reviewer often yields comparable or better results:
>
> 🔗 https://anonymous.4open.science/r/16368/Performance%20of%20combinations.pdf
>
> Regarding runtime, we now report detailed timings using violin plots:
>
> 🔗 https://anonymous.4open.science/r/16368/Runtime%20violin%20plot.pdf
>
> - **Median runtime per MR** is **6.2 minutes**.
> - The overall CI/CD pipeline (including compilation, analysis, and deployment checks) typically takes 15–30 minutes.
> - Our module runs **in parallel** from the beginning and does **not introduce blocking delays**.
>
> Thus, we believe the overhead is acceptable in practical scenarios. Further analysis will be added to **Appendix M**.
>
> ---
>
> ### **4. Prompting Strategy and Retrieval-Augmented Generation**
>
> **(Response to “Weakness 3”)**
>
> > In the LLM prompts, the authors can discuss the effectiveness of Retrieval-Augmented Generation...
>
> Thank you for raising this point. Although we do not use explicit RAG pipelines, our **slicing mechanism** serves a similar purpose: retrieving and providing only **relevant context slices** to the model. This is evaluated in **Section 5.2 (RQ2)** and will be clarified as RAG-aligned in the revision.
>
> ---
>
> ### **5. Minor Corrections and Reference Additions**
>
> - The incorrect reference to Section 2.1 in **Appendix A** will be fixed.
> - We will include the three suggested references and briefly discuss their relation to our work.
>
> ---
>
> **We hope this response addresses your concerns and supports a more favorable assessment. Your feedback has helped us strengthen the clarity and rigor of our work.**

---

### Decision · Program_Chairs · 2025-05-01

**Decision:**

Accept (spotlight poster)

**Comment:**

The reviewers appreciated that the work was tested in practically realistic setting and combines multiple techniques into a strong-performing tool. The paper is well-written and the authors made prompt and thorough improvements based on reviewer feedback. Given this, we recommend the paper for acceptance.